# The Neuroprotective Effects and Therapeutic Potential of the Chalcone Cardamonin for Alzheimer’s Disease

**DOI:** 10.3390/brainsci13010145

**Published:** 2023-01-14

**Authors:** Kimberly Barber, Patricia Mendonca, Karam F. A. Soliman

**Affiliations:** 1Division of Pharmaceutical Sciences, College of Pharmacy and Pharmaceutical Sciences, Institute of Public Health, Florida A&M University, Tallahassee, FL 32307, USA; 2Department of Biology, College of Science and Technology, Florida A&M University, Tallahassee, FL 32307, USA

**Keywords:** cardamonin, neuroprotection, anti-inflammatory, antioxidant, neurodegeneration

## Abstract

Neurodegenerative diseases (ND) include a wide range of conditions that result from progressive damage to the neurons. Alzheimer’s disease (AD) is one of the most common NDs, and neuroinflammation and oxidative stress (OS) are the major factors in the development and progression of the disease. Many naturally occurring phytochemical compounds exhibit antioxidant and anti-inflammatory activities with potential neuroprotective effects. Several plant species, including *Alpinia katsumadai* and *Alpinia conchigera*, contain cardamonin (CD). CD (2′,4′-dihydroxy-6′methoxychalcone) has many therapeutic properties, including anticancer, anti-inflammatory, antioxidant, antiviral, and antibiotic activities. CD is a potent compound that can reduce OS and modulate the inflammatory processes that play a significant part in developing neurodegenerative diseases. CD has been shown to modulate a variety of signaling molecules involved in the development and progression of ND, including transcription factors (NF-kB and STAT3), cytokines (TNF-α, IL-1, and IL-6), enzymes (COX-2, MMP-9, and ALDH1), and other proteins and genes (Bcl-2, XIAP, and cyclin D1). Additionally, CD effectively modulates miRNA levels and autophagy-related CD-protective mechanisms against neurodegeneration. In summary, this review provides mechanistic insights into CD’s ability to modify multiple oxidative stress–antioxidant system pathways, Nrf2, and neuroinflammation. Additionally, it points to the possible therapeutic potential and preventive utilization of CD in neurodegenerative diseases, most specifically AD.

## 1. Introduction

Neurodegenerative diseases (NDs) are a diverse category of conditions characterized by the loss of neurons over time, with debilitating illnesses that cause gradual degeneration or neuronal death. Alzheimer’s (AD) and Parkinson’s disease (PD), amyotrophic lateral sclerosis (ALS), frontotemporal dementia (FTD), Huntington’s disease (HD), and prion disease are examples of this group of diseases. The two most frequent NDs are AD and PD. According to the AD Association 2021 report, more than 6.2 million Americans could suffer from AD [1]. PD is expected to affect 1.2 million persons in the US by 2030 [2].

NDs have become more prevalent as the population ages. However, the mechanisms that direct the destabilization of synapses and lead to neuron death remain obscure. Although there are no available treatments to cure neurodegenerative conditions, there is an increasing range of therapeutic, primarily symptomatic and supportive options. Recently, there has been a surge in interest in naturally occurring phytochemical compounds with antioxidant, anti-inflammatory, and neuroprotective potential over the last few decades, not only because of concerns about the side effects of conventional medicine but also because natural products are typically inexpensive and readily available in an ingestible form [3]. Furthermore, medicinal plants have been used as effective treatments for various illnesses and disorders for thousands of years since they are safe and accessible [4].

Cardamonin (CD) belongs to the chalcones, natural organic compounds that have been described as having various biological activities such as antioxidant, cytotoxic, and anticancer properties. CD is a 2′,4′-dihydroxy-6′-methoxychalcone found in high concentration in the *Alpinia katsumadai* seeds [4], has antinociceptive, anti-inflammatory, antiprotozoal, antiulcer, antihistaminic, and anti-tumor properties while inhibiting vascular smooth muscle cell proliferation and migration [5]. CD protects against cisplatin-induced nephrotoxicity [6] and reduces the proliferation and migration of vascular smooth muscle cells [7]. CD effects have been suggested by inhibiting the redox-activated NF-kB and cytokines [7]. Because of its antioxidant, anti-inflammatory, and neuroprotective properties, CD was shown to inhibit the damaging effects of oxidative stress (OS) and neuroinflammation that play a key role in the pathogenesis of NDs, such as AD [8]. In this review, we will examine the neuroprotective effects of CD, emphasizing the anti-neuroinflammatory and anti-OS effects, which can lead to the prevention of NDs, in particular AD.

## 2. Methods for Literature Review

This narrative review was designed to provide an overview of the publications that describe the therapeutic role of CD on neurodegeneration through several molecular mechanisms. Main key words were used to search relevant papers that investigated the pharmacological properties of CD. The PubMed database was used to find papers published since 1995 that describe scientific research on CD vs. neurodegeneration, neuroprotection, neuroinflammation, cytokines, anti-inflammatory, and antioxidant effects, AD, PD, Nrf2, microglia, MAPK and NF-kB signaling, autophagy, and micro-RNA. The conclusion was based on the screening of data from these papers.

## 3. Neurodegenerative Diseases

### Alzheimer’s Disease Pathogenesis, Risk Factors, and Oxidative Stress

According to the World Alzheimer Report of 2021, around 50 million people worldwide have dementia. This number is predicted to rise to about 82 million in 2030 and 152 million in 2050. There are approximately 200 types of dementia, with AD accounting for 50–60% of all cases [9]. Alois Alzheimer first defined this type of dementia in Frankfurt in 1907, and it is the most common ND [10]. During the COVID-19 pandemic, AD increased by 16% in the United States. AD kills more people than both breast and prostate cancer combined. The economic impact of AD and other dementias is also substantial, and by 2022 can be expected to cost the United States $355 billion and reach $1.1 trillion by 2050. In contrast, deaths from heart disease declined by 7.3% between 2000 and 2019, whereas deaths attributed to AD climbed by 145%. Because of this vast impact, it is vital to develop effective therapies for AD prevention and/or slowing its progression [11].

Even though numerous studies concerning AD pathophysiology have been conducted during the past few years, AD’s origin is still unknown. Dementia is mostly a sporadic, age-related disease, with hereditary alterations accounting for less than 5% of all cases [12]. Between the ages of 65 and 85, the risk of developing AD increases 14-fold, and nearly 47% of people over 85 are affected. The continuous decline in cognitive abilities is a hallmark of this disease. Short-term memory impairment, which interferes with and complicates daily activities, together with impairment in many cognitive domains, such as logical understanding, language, orientation, behavior, judgment, and motor problems. This development is connected to a considerable decrease in brain volume in these patients [13], which is caused by the loss of neurons and the degradation of synapses, especially in the hippocampus, which is crucial for memory and spatial orientation [14].

The presence of amyloid plaques in the extracellular space of AD brains is one of the disease’s hallmarks. They contain agglutinated peptides amyloid-β (mostly Aβ1–40/42) that have β sheet structures and form fibrils [15]. The disruption of amyloid-peptide homeostasis mediated by proteolytic destruction of the amyloid precursor protein (APP) causes amyloid-peptide buildup [15]. In addition, hyperphosphorylated tau protein, neurofibrillary tangles, synapse loss, dystrophic neurites, and substantial gliosis have been found in AD brains [16]. The genesis of this degenerative brain impairment is complicated by several factors. AD etiology may include age, environmental factors such as chronic stress, traumatic brain injury, internal processes such as chronic pain, OS, inflammation, genetic factors such as mutations in the genes encoding the APP, preselinin-1 and -2, and the apolipoprotein E (ApoE) [17].

AD is a prevalent neurological illness that progresses over time. Cognitive and memory loss are common in AD patients. Extracellular Aβ deposits, in the form of senile plaques and intracellular neurofibrillary tangles (NFT) formed of helical-coupled tau protein, cause cerebral degeneration with selective neuronal death. Glial cell activation in the brains of AD patients enhances the generation of excessive amounts of free radicals, nitric oxide, and cytokines, all of which may be harmful to neuronal cells [18]. Synaptic damage and neuronal death are hallmarks of the neurodegenerative process of AD. According to recent data, alterations in adult neurogenesis in the hippocampus may also have a role. Synaptic loss is one of the most significant predictors of cognitive decline in AD patients. In AD, synaptic dysfunction and faulty neurogenesis are linked to the accumulation of Aβ, an imbalance between the levels of Aβ production, aggregation, and clearance results in the creation of hazardous oligomers [19].

Many of the risk factors related to dementia in general and AD, in particular, have been recognized in previous reviews. It was discussed that over 20 different risk factors are linked to AD. It proposed that common pathological changes associated with the disease are amyloid immunoreactive senile plaques and tau immunoreactive NFT [20]. As a result, over 60 environmental risk variables were discovered and grouped into six categories in a recent assessment: air quality, heavy metals, other metals, trace elements, occupational exposure, and miscellaneous [21]. It is still unknown how many seemingly unrelated risk factors can lead to AD [20]. Previous assessments have found that AD is associated with many potentially unrelated risk factors. Many of the hallmarks of AD pathology can be seen to some extent in the aging but cognitively normal brain [22].

Various antioxidant mechanisms balance the production of reactive oxygen species (ROS) in a healthy state. OS occurs when the formation of ROS and antioxidant defenses are out of balance, resulting in an excessive buildup of ROS. Damage to cell membranes from lipid peroxidation, protein structure and function changes due to protein oxidation, and structural damage to DNA are all possible causes of OS [23].

The brain is particularly prone to OS since it is one of the most metabolically active organs in the body. First, the brain has a high oxygen demand, accounting for 20% of total oxygen consumption. Second, redox-active metals, such as iron and copper, are plentiful in the brain and are involved in catalyzing ROS production. Third, the membranes of brain cells contain significant quantities of polyunsaturated fatty acids, which act as lipid peroxidation substrates. Fourth, glutathione (GSH) levels in the brain are low, even though it acts as an endogenous antioxidant in removing ROS [24]. In AD, it has been proposed that oxidative imbalance and the resulting neuronal damage play a key role in the onset and progression of the disease [25]. However, increased ROS generation and the mechanisms underlining the disruption of redox balance in individuals with AD may cause mitochondrial dysfunction [26].

Even in the early stages of AD, the accumulation of Aβ appears to enhance OS, leading to mitochondrial malfunction and energy failure [27]. Previous research has suggested that Aβ-induced oxidative imbalance may raise the levels of byproducts of lipid peroxidation (e.g., 4-hydroxynonal, malondialdehyde), protein oxidation (e.g., carbonyl), and DNA/RNA oxidation (e.g., 8-hydroxyldeoxyguanosine and 8-hydroxylguanosine). Patients with AD, on the other hand, have lower amounts of antioxidants (such as uric acid, vitamin C, and E), as well as antioxidant enzymes (such as superoxide dismutase, catalase, and others) [25,26]. Furthermore, AD transgenic mice expressing mutant APP and presenilin-1 showed higher levels of H_2_O_2_ and protein and lipid peroxidation, showing that Aβ may exacerbate OS in AD [26,28]. OS has been shown to stimulate the synthesis of Aβ in several earlier investigations. Defects in antioxidant defense pathways were found to generate increased OS, which encouraged Aβ depositions in transgenic mice with the APP mutation [26].

## 4. Down-Regulation of Nuclear Factor Erythroid 2-Related Factor 2 (Nrf2) Antioxidant Activity in the Elderly

Internal metabolism and environmental exposure constantly generate ROS and reactive nitrogen species (RNS) in the body. Reactive oxidants are produced in normal cells in a controlled manner, and some of them are beneficial. Cell division, inflammation, immunological function, autophagy, and stress response are all regulated by oxidants generated in response to physiological signals, which operate as essential signaling molecules [29]. Uncontrolled oxidant production disrupts cellular processes and contributes to cancer, chronic illness, and toxicity. Reactive oxidants appear to be crucial regulators of both physiological and pathological outcomes from prokaryotes to humans [29].

The nuclear factor erythroid 2-related factor 2 (Nrf2) is an essential region leucine zipper (bZip) transcription factor that belongs to the cap ‘n’ collar (CNC) subfamily. Its binding motif, a cis-regulatory sequence in the globin locus control area required for erythropoiesis and platelet formation, was used to clone Nrf2 [30]. Although Nrf2 does not appear to be needed for blood cell development, it has been discovered to mediate the induction of several drug-metabolizing enzymes (DMEs) by antioxidants and electrophiles, including glutathione S-transferase (GST) and NAD(P)H: quinone oxidoreductase 1 (NQO1) [29].

The antioxidant response element (ARE), a transcriptional regulatory element, controls the dynamic expression of phase II genes, which resembles the Nrf2-binding motif. The major transcription factor, Nrf2, works on these AREs (cis-acting elements), boosting the production of an array of genes that recover the redox insult caused by chemicals and xenobiotics, ultimately protecting the cells from damage [31]. An increase in OS, a common feature of aging, has been linked to several age-related diseases. When people get older, their bodies produce more oxidants from various sources, but their antioxidant enzymes, which are the body’s first line of defense, decline. Repair mechanisms, such as proteasomal breakdown of damaged proteins, are also deteriorating. Significantly, as people age, their adaptive response to OS decreases. Many antioxidant enzymes, including the proteasome, are regulated by nuclear factor erythroid 2/electrophile-responsive element (Nrf2/EpRE) signaling, which is both basal and inducible. The activity of Nrf2/EpRE is influenced by several factors, including transcription, post-translation, and interactions with other proteins [32].

Explaining the signaling route by which such responses are regulated has been the most significant finding in recent decades. The activation of Nrf2 and its interaction with Kelch-like ECH-associated protein 1, Keap1, is crucial to our knowledge of such regulation [32]. The transcription factor Nrf2 regulates the expression of various antioxidant and detoxification enzymes. Due to several interacting proteins and regulatory molecules, the Nrf2 signaling system has emerged as an essential cellular defense and survival route against OS and toxicants. In humans and model organisms, disruption of Nrf2 signaling is linked to a higher vulnerability to oxidative insults and other toxins [32]. With age, the tightly regulated nucleophilic tone is weakened, resulting in a chronic oxidative state in old organisms. Age-related oxidative damage is associated with an increase in oxidant production, a decrease in antioxidant capacity, and less efficient activity of both the proteasome and the mitochondrial Lon protease, resulting in the accumulation of oxidized and cross-linked protein aggregates in intracellular and intramitochondrial masses [32]. According to mounting evidence, the loss of adaptive antioxidant responses to oxidative stimuli, particularly the Nrf2/EpRE signaling system, is also a critical factor in the accumulation of oxidative damage in aging [32].

Age affects Nrf2 and ARE signaling, although there is conflicting evidence that aging causes changes in the expression of Nrf2 target genes. ARE–human placental alkaline phosphatase (hPAP) reporter mice, on the other hand, appear to have increased Nrf2 activity in the dorsal horn of the spinal cord with age, but low-levels of ARE activity is evident in 60-day-old mice, according to data compiled from many independent studies [33]. High levels of ARE activity can be seen in mice as young as 110 days old and in ARE–hPAP animals as young as 6 months old. With age, ARE activity emerges in the ventral horn of the spinal cord of these mice, however, to a lesser extent [33]. This is supported by increased Nrf2 protein in the spinal cord and enhanced Nrf2 activity in the cerebellum of elderly mice. The substantia nigra, ventral tegmental region, and hippocampus of old rats had lower levels of Nrf2 and gene expression compared to the spinal cord and cerebellum. ARE activity appears to be highest in the substantia nigra, striatum, and ventral hippocampus in juvenile ARE–hPAP animals. As a result, it seems that the Nrf2 response with age varies by region and that aging is related to a reduction in Nrf2 activity, particularly in regions where Nrf2 activity is elevated in young animals [33]. The involvement of Nrf2 in aging in many cell types has received little attention. Even though ARE activity can be found in a range of cell types, including those in the spinal cord, the increased Nrf2 protein detected in the spinal cord of aged mice is primarily located in astrocytes [33]. Moreover, several studies show that the ability of Nrf2 to respond to insults changes with age. In rhesus macaque vascular smooth muscle, the Nrf2 response to injury is reduced in older animals relative to younger animals. In *Drosophila*, the response of the Nrf2 homolog CncC to an acute injury is reduced in older flies relative to younger flies. In the cerebellum of young mice, exposure to airborne pollution evokes a significant Nrf2 response, whereas the same treatment in older mice had little effect on Nrf2 signaling [33]. Treatment of older rats with testosterone restores Nrf2 function in the basal ganglia but not the hippocampus and is linked to enhanced exploratory and motor behavior and increased dopaminergic cell counts. These findings imply that Nrf2 ability to respond to insults changes with age and is region-dependent [33].

In the aging rat liver, GSH levels drop dramatically. The levels and activity of gamma-glutamylcysteine ligase (GCL), the rate-controlling enzyme in GSH synthesis, were evaluated because GSH levels are partly a reflection of its synthetic capability [34]. The GCL’s catalytic (GCLC) and modulatory (GCLM) subunits were reduced by 47% and 52% with age, respectively. GCL activity decreased by 53% in parallel with reducing subunit levels. It was hypothesized that aging resulted in the dysregulation of Nrf2-mediated GCL expression because Nrf2 regulates baseline and inducible GCLC and GCLM expression via the ARE. A 50% reduction in total and nuclear Nrf2 levels was found with age, implying a decrease in Nrf2-dependent gene transcription. In addition, to investigate if the inducible nature of Nrf2 nuclear translocation is affected by the constitutive loss of Nrf2 transcriptional activity, aged rats were given R-(α)-lipoic acid (LA), which is a disulfide molecule that has been found to activate Nrf2 in vitro and raise GSH levels in vivo [34]. After 12 h, LA treatment boosted nuclear Nrf2 levels in aged rats. LA also caused Nrf2 to bind to the ARE, resulting in greater GCLC levels and GCL activity 24 h after LA injection. As a result, age-related GSH synthesis loss could be driven by the deregulation of ARE-mediated gene expression, while chemoprotective drugs such as LA can help to mitigate this loss [34].

## 5. Biomarkers for Diagnosis and Treatment of Alzheimer’s Disease

It is becoming more crucial than ever to identify biomarkers for the diagnosis of AD. To date, the most advanced approach to detect AD with high specificity and sensitivity is the ELISA assessment of β-amyloid (1–42), total tau, and phospho-tau-181 in cerebrospinal fluid (CSF) [35].

Throughout the pathophysiological progression of AD, glial cells maintain an inflammatory response that is overexpressed and synergizes with amyloid-β and tau buildup, inducing synaptotoxicity and neurodegeneration. Clinical trials with anti-inflammatory drugs, such as non-steroidal anti-inflammatory drugs (NSAIDs), failed to meet primary efficacy endpoints despite a compelling therapeutic justification. It is possible that study design difficulties, including deficiency of diagnostic accuracy and new biomarkers to identify the target group and provide the process, contributed to the disappointing results [36]. However, according to a recent meta-analysis, NSAIDs may have a biological property. Candidate fluid biomarkers of neuroinflammation, such as triggering receptors expressed on TREM2, IL-1β, MCP-1, IL-6, TNF-α receptor complexes, TGF-β, and YKL-40, are undergoing analytical/clinical validation in this area. To conduct in vivo and longitudinal regional explorations of neuroinflammation, positron emission tomography (PET) radioligands are also being studied. Biomarkers that track various biochemical pathways (body fluid matrixes) and brain neuroinflammatory endophenotypes (neuroimaging markers) can help to unravel the temporal-spatial dynamics of neuroinflammation and other AD pathophysiological mechanisms. Large-scale clinical trials that directly or indirectly investigate new-generation drugs active on neuroinflammatory targets and demonstrate putative disease-modifying effects are likely to benefit from robust biomarker–drug development pipelines [36].

### 5.1. The Role of Neuroinflammation in Alzheimer’s Disease

Neuroinflammation begins decades before the clinical onset of AD and is one of the earliest changes during AD development. Several genetic variants—TREM2, CD33, PILRA, CR1, MS4A, CLU, ABCA7, EPHA1, and HLA-DRB5-HLA-DRB1—have been related to neuroinflammation in genome-wide association studies. These genes are associated with cytokines/interleukins/cell turnover, pro-inflammatory intracellular signaling, lipid metabolism, synaptic activity, and vesicle trafficking. Neuroinflammation is thought to be caused by a slew of interrelated abnormal molecular pathways triggered and maintained by TNF-α, TGF-β, IL-1β, and the triggering receptor protein TREM2. Microglia and astrocytes are essential neuroinflammatory drivers and regulators. They are necessary for neurotransmission and synaptic homeostasis in physiological settings [36].

AD is characterized by neuroinflammation, and microglia (the brain’s resident phagocyte) play an important function in the immunological response seen in this condition. Microglia serve as sentinels and protectors, but in AD, they may become overly reactive, causing neuropathology. Furthermore, a mutation of low-frequency in the gene encoding the TREM2 has recently been discovered to confer a higher risk of AD during late-onset (LOAD) cohorts with an effect size similar to that of APOE previously the single genetic risk factor related to LOAD [37].

The key neuropathological characteristics of AD are amyloid plaques and NFT; however, it is becoming increasingly clear that neuroinflammation plays an important function in the etiology of the disease. The usual function of neuroinflammation is to defend the CNS from infectious insults, damage, or disease by activating the innate immune system in the brain [38]. Neuroinflammation has long been known to play a role in neurological diseases and AD [39]. It is an intricate response that includes a slew of cellular and molecular modifications, recruitment of peripheral immune cells, intracellular signaling pathway stimulations, and the discharge of inflammatory mediators in the brain. All these factors, alone or in combination, can cause neuronal malfunction and death in AD [40,41]. According to these studies, neuroinflammation is an early and persistent characteristic of AD.

Increasing data reveals that the pathophysiology of AD is not limited to the neuronal compartment but also involves immune systems in the brain [42]. Misfolded and aggregated proteins link to pattern recognition receptors on micro- and astroglial cells, triggering an innate immune response marked by the release of inflammatory mediators that contribute to disease progression and severity [42]. According to a genome-wide investigation, some genes are linked to sporadic AD, coding for components that control glial clearance of misfolded proteins and the inflammatory response [42]. Understanding and managing these interactions could be the key to preventing or delaying the onset of many late-onset CNS illnesses. In this scenario, neuroinflammation plays an equal or greater role in AD etiology than the plaques and tangles [42].

AD is a neurological illness that affects individuals as they age. AD prevalence increases with age with the progressive augmented chronic low-grade inflammation (inflammaging) that may contribute to the ND process in AD [43]. Although the specific mechanisms are unknown, abnormally high amounts of reactive oxygen and nitrogen species (RONS) in the brain caused by various endogenous and external processes may alter cell communication and provoke cellular senescence, inflammation, and pyroptosis. Furthermore, a weakened immune privilege of the brain that allows peripheral immune cells and infectious pathogens to infiltrate the brain could play a role [43]. Meta-inflammation and gut microbiota dysbiosis may also play a role in the neuroinflammatory process, given that inflammatory/immune pathways are dysregulated alongside cognitive failure in AD. Understanding the link between the CNS and the immune system may aid in the development of safe and effective therapeutics for the disease. Chronic sterile low-grade inflammation or inflammaging is associated with aging, including cellular senescence, immunosenescence, mitochondrial dysfunction, defective autophagy, and mitophagy, dysregulation of the ubiquitin–proteasome system, activation of the DNA damage response, and meta-inflammation or meta-inflammation from chronic overnutrition or obesity [43]. These processes induce changes in circulating immune markers, such as, IL-6, C-reactive protein, TNF-α and its receptors (tumor necrosis factor receptor I II), d-dimer, vascular cell adhesion molecule I, and sirtuin signaling. Worldwide, there are currently 728 million people who are 65 years or older. This number is predicted to double in the following 30 years, surpassing 1.5 billion in 2050. As a result, the number of individuals susceptible to inflammation will rise dramatically in the coming decades [43].

While there is clear evidence that AD has a complex etiology, neuroinflammation plays a main function in its pathogenesis because of its ability to worsen Aβ-related disorders. PET in AD patients’ brains shows increased microglia activation (inflammation). Aβ may stimulate the brain’s innate immune cells, and pro-inflammatory cytokines levels are higher in AD patient serum and postmortem brain [43]. Microglia, astrocytes, mast cells, oligodendrocytes, chemokines, cytokines, and complement have a role in the prolonged inflammatory response in AD brains that goes beyond a reaction to neuronal loss. These factors contribute to the disease’s setup and progression [43].

### 5.2. Microglia Role in Neuroinflammation

Various proteins that develop during ND, such as beta-amyloid, tau, heat shock proteins, and chromogranin, act as danger-associated molecular patterns that activate inflammatory signaling when recognized by pattern recognition receptor pathways and eventually result in the production and release of immune mediators. These may initially have positive benefits, but they ultimately impede neural function and lead to cell death [44]. The microglia surrounding plaques are indeed defective at Aβ uptake in cortical tissue specimens from patients with AD. Radioligands are used in new PET techniques to detect active microglia in vivo. Many tracers are directed towards the 18 kDa translocator protein (TSPO), a mitochondrial outer membrane protein found in microglia that is increased during activation. Microglia are phagocytic cells that can consume Aβ via various cell surface receptors, including a cluster of differentiation CD-14, Toll-like receptor (TLR)-2, TLR4, integrin alpha-6/beta-1 (ITGA6:ITGB1), CD47, and scavenger receptors such as CD36 [44]. Failure of microglia to eliminate extracellular amyloid has been considered a crucial element in the accumulation of Aβ throughout the brain in AD [44].

Most inflammatory stimuli cause glial cells, such as astrocytes and microglia, to produce cytokines [44]. Many cytokines have been linked to advanced AD pathology, including IL-1β and IL-12. Increased IL-1β serum levels have been associated with AD in moderate cognitive impairment patients [44]. IL-1β polymorphisms and the start of AD pathology have been linked in several studies, with both IL-1β polymorphisms and APOE-4 being linked to greater levels of IL-1β in the blood and sleep disturbances in patients [44]. An IL-12 polymorphism has been linked to AD in a Han Chinese community, and it is involved in the control of the adaptive and innate immune systems [44]. In APP/PS1 mice, ablation of the anti-inflammatory cytokine IL-10 reduced AD-related impairments such as decreased synaptic integrity and behavioral abnormalities. In the APP transgenic mice model, Chakrabarty et al. found that overexpression of IL-10 using adeno-associated viruses (AAVs) increased amyloid deposition, behavioral impairments, synaptic changes, and decreased microglial phagocytosis of Aβ [44].

Microglia also play a role in the protection and remodeling of synapses, which is necessary for the correct maintenance and plasticity of neural circuits [42]. This action is mediated, to some extent, by releasing trophic factors, such as brain-derived neurotrophic factor, which aids memory formation. Microglia extend their processes to the injury site after being activated by pathogenic stimuli such as neuronal death or protein aggregation, later migrating to the lesion and initiating an innate immune response [42]. Microglia in AD can bind to soluble Aβ oligomers and fibrils via receptors such as class A scavenger receptor A1, CD36, CD14, 6-1 integrin, CD47, and Toll-like receptors (TLR2, TLR4, TLR6, and TLR9), which is assumed to be a part of the inflammatory response in the disease. The Aβ peptide is made up of two membrane-bound proteases that cleave a bigger precursor, the APP; the β-site APP cleaving enzyme 1 (BACE1) protease is responsible for the initial cleavage, which is followed by an unusual cleavage by the secretase complex within the transmembrane region of APP, resulting in variably truncated C-termini ranging from 37 to 42 amino acids [42]. The 42 amino acid long version of Aβ has a considerable proclivity for forming soluble oligomers and fibrils. When Aβ binds to CD36, TLR4, and TLR6, microglia become activated and release pro-inflammatory cytokines and chemokines. In vitro genetic ablation of CD36, TLR4, or TLR6 lowers Aβ-induced cytokine production while preventing intracellular amyloid buildup and inflammasome activation [42].

### 5.3. Intracellular Signaling Pathways Activated in Neuroinflammation

Intracellular signaling pathways, such as the mitogen-activated protein kinase (MAPK) and nuclear factor-kappa B (NF-kB) signaling pathways, modulate inflammatory cytokine production in activated microglia and astrocytes [45].

#### 5.3.1. MAPK Signaling in Neuroinflammation

MAPKs are a group of serine/threonine protein kinases that play an essential role in the generation of inflammatory mediators. Extracellular signal-regulated kinases (ERK 1/2), c-Jun N-terminal kinase (JNK), and p38 isoforms are some of the most critical kinases in this group. Increased p38 MAPK activity has been linked to pathologies associated with ND disorders [45]. The p38 MAPK is increased during microglial inflammation, according to several studies. According to evidence, patients with neurological disorders, including AD, may benefit from p38 MAPK inhibitors. As a result, medications that interfere with these kinases’ activity are needed to modify age-related and activated microglia-associated neuroinflammatory disorders [45].

The MAP-ERK pathway is a critical component of the neuroinflammatory pathway triggered by glial cells during the progression of ND disorders such as PD, AD, Huntington’s disease, amyotrophic lateral sclerosis, and preonic diseases [46]. The mechanism induced by MAPK/ERK activation differs depending on the developmental stage (mature or senescence), the type of cellular element in which the pathway is activated, and the anatomical brain structure. However, several of these pathways have significant gaps in the phosphorylated ERK targets [46].

MAPK and NF-KB are two intracellular signaling molecules that regulate enzymes involved in the inflammatory process. Cyclooxygenases and prostaglandin (PG) E synthases (PGESs) catalyze the formation of PGE2, an AA derivative [45]. There are two types of cyclooxygenases: cyclooxygenase-1 (COX-1) and cyclooxygenase-2 (COX-2). In cultured microglia, the bacterial cell wall component, lipopolysaccharide (LPS), has been found to overexpress COX-2 [45]. PGESs control the penultimate stage in the production of PGE2. Three PGESs have been identified so far: microsomal PGESs, microsomal prostaglandin E synthase-1 (mPGES-1), and cytosolic PGESs (cPGES). mPGES-1 is an inducible enzyme that has been demonstrated to be increased in activated microglia [45].

#### 5.3.2. NF-kB Signaling in Neuroinflammation

NF-kB is a transcriptional regulator that modulates cellular biological activity by attaching to a promoter region in the nucleus and transcribing different protein genes. A recent study has linked NF-kB to autoimmune disorders, inflammatory diseases, cardiovascular diseases, and ND diseases [47]. As a result, targeting the NF-kB protein as a therapeutic method opens up new possibilities. The activation of the IΚΒ kinase/NF-kB signaling pathway causes a variety of clinical illnesses in humans, including ND, inflammatory, autoimmune, and cancer diseases. As a result, IΚΒ kinase/NF-kB transcriptional activity is tightly controlled through many cascade pathways. The NF-kB pathway regulates the expression of pro-inflammatory genes such as cytokines, chemokines, and adhesion molecules [47]. In response to various stimuli, the cytosolic sequestered NF-kB is phosphorylated and translocated into the nucleus, where it further transcribes numerous genes necessary for changing various cellular activities through interacting with an inhibitor molecule protein (Iκβ). The role of different NF-kB family member proteins in expressing diverse gene products and mediating numerous cellular cascades has been proven in several studies. MicroRNAs serve a critical function in regulating the inflammatory process as regulators of NF-kB. As a result, NF-kB inhibitors and their family members represent a unique therapeutic target in the prevention of various disorders. The NF-kB signaling pathway may be regulated as a safe and effective therapy option for various illnesses [47].

An N-terminal Rel homology domain (RHD) in NF-kB proteins supports dimerization and binding to kB sites by establishing contact with DNA. In mammals, there are five members of the NF-kB family: RelA/p65, RelB, c-Rel, p50 (NF-kB1), and p52 (NF-kB2). P65, c-Rel, and RelB are the only members of this family with C-terminal transactivation domains (TADs), which increase transcription initiation. The signaling pathways for NF-kB have been divided into canonical (classical) and noncanonical. Activating stimuli, such as cytokines, when recognized by receptors, such as TNFR, IL-1 receptor (IL-1R), TLRs, and antigen receptors, start signaling cascades that eventually activate Iκβ (inhibitor of κβ) kinase (IKKβ). Activated IKKβ phosphorylates Iκβ proteins, allowing them to be ubiquitinated and degraded by proteasomes [48]. Some of the cytokine family members activate IKKβ, leading to the phosphorylation of p100 and the formation of p52/RelB complexes in the noncanonical NF-kB pathway [48]. Notably, both IKKα and IKKβ can interact with other signaling pathways including the p53, MAP kinase, and IRF pathways to control transcriptional changes directly (Figure 1). NF-kB has been demonstrated to regulate the transcription of several genes involved in pro-inflammatory, inflammatory, stress, and growth responses [48]. Indeed, transcriptional modulation of cytokine production, such as TNF-α and IL-1α/β, and proteins involved in antigen presentation, such as MHC class I and 2 microglobulin, is one of the well-studied roles of NF-kB in aging and many diseases. NF-kB also controls the expression of chemokines, such as MCP-1 and MIP-1, as well as adhesion molecules, such as ICAM-1, E-selectin, and VCAM-1, to aid immune cell recruitment and attachment at inflammatory sites [48]. To increase cell proliferation and survival, NF-kB modulates pro-apoptotic factors, including Bim and Bax, anti-apoptotic factors, such as XIAP and BCL-2, as well as growth factors including nerve growth and vascular endothelial growth factors [48].

### 5.4. Cytokines and Neuroinflammation

Cytokines are nonstructural proteins released by various cells with a variety of biological functions. One of the key mechanisms of AD neuropathology is chronic neuroinflammation generated by cytokines secreted by activated microglia and astrocytes [39]. TNF-α is a pleiotropic cytokine that regulates various physiological and pathological processes, including cell death, differentiation, and inflammation. Under physiological settings, TNF-α expression levels in the healthy brain are modest [49]. TNF-α is considered necessary in inflammatory and disease states, being identified as a neuromodulator in the progression of AD symptoms [50]. TNF-α production dysregulations have been associated with AD [51]. It has been demonstrated to modulate APP in a mouse model of AD131 and increase Aβ formation by boosting BACE1132 expression and secretase activity [52]. The TNF-α synthesis inhibitor (3,6′-dithiothalidomide) reduces inflammation and microglial activation produced by Aβ, neuronal degradation, and memory impairment [53].

IL-1 is an inflammatory cytokine produced mainly through microglia in the brain, and it appears to have a function in the etiology of AD [54]. A 10-year follow-up study of older adults in a nursing home showed that systemic inflammatory IL-1 levels during the agitation stage are risk factors for the development of AD [55]. By regulating secretase activity in neurons, IL-1β can enhance Aβ synthesis [52]. Increased APP production in astrocytes is aided by improved IL-1 message translation [56]. In the culture medium of primary neurons, IL-1 raised the amounts of soluble amyloid precursor protein (sAPP) in a dose-dependent manner. IL-1 also causes phosphorylation of tau proteins and paired-helical filament production, which aggregate into NFTs, according to certain studies. In an IL-1 overexpression (IL-1XAT) AD mice model sustained IL-1 overexpression increased tau phosphorylation and exacerbated tau pathology [32].

Long-term memory is similarly harmed by chronic hippocampal IL-1 overexpression and injections into the peripheral and intracerebral circulation. However, one study suggests that IL-1 may help to reduce the progression of AD. IL-1 overexpression led to plaque clearance regardless of cytokine expression length or age, and sustained IL-1 did not induce obvious apoptosis in the hippocampus of an AD mice model. Therefore, there is no question that IL-1 promotes AD abnormalities in neuroinflammation, but its mechanism is unknown, and several crucial questions remain unanswered [32].

In the CNS, IL-6 is a significant cytokine with a considerable impact on the brain. The IL-6 gene corresponds to chromosome p21 and is a promising candidate for a genetic risk factor for AD [57]. The IL-6 gene polymorphisms are related to AD risk [58,59,60]. IL-6 stimulates and increases the migration of microglia and astrocytes in the AD brain, causing them to produce pro-inflammatory cytokines and promoting tau phosphorylation in neurons [61]. However, a meta-analysis found that polymorphism in the IL-6-174G/C gene offers protection for AD in Asians but not Caucasians. It may help to reduce the risk of AD [62].

One of the essential anti-inflammatory cytokines, IL-10, is involved in cell survival and neural homeostasis. The generation of pro-inflammatory cytokines caused by Aβ or LPS is inhibited when glial cells are pre-exposed to IL-10 [63]. The activation of IL-10, on the other hand, had no effect on Aβ deposition in the mouse brain. It was ineffective at degrading Aβ42, and it may raise the Aβ42:Aβ40 ratio in Tg2576 mice (Aβ42 promotes amyloid formation while Aβ40 prevents it). More data is needed to support the link between IL-10 and AD [64].

### 5.5. The Association between NF-kB and Nrf2

Recent research has found that the Nrf2 and NF-kB signaling pathways interact under stress and in many pathological circumstances. Nrf2 deficiency is linked to increased inflammation, whereas its overexpression reduces pro-inflammatory and immunological responses that NF-kB transcriptionally regulates. Innamorato et al. found that Nrf2 knockout mice were hypersensitive to LPS-induced inflammation, as evidenced by increased inflammatory markers, such as F4/80 (both mRNA and protein expression), inducible NO synthase, IL-6, and TNF-α, compared to wild-type littermates’ hippocampi [65]. Surprisingly, treatment with sulforaphane, a chemical that increases Nrf2-dependent gene expression, prevented these detrimental alterations [48]. Additional studies using other cell types, such as microglial cells and monocytes, have backed up this evidence.

Furthermore, irrespective of its redox regulatory action, Kobayashi et al. demonstrated that Nrf2 can inhibit the transcriptional activation of pro-inflammatory genes. In human monocytes, silencing TNF-α increased Nrf2 and acute inflammatory responses mediated by NF-kB. However, prolonged activation of Nrf2 by TNF-α resulted in autocrine regulation of the protein, implying that the interplay between Nrf2- and NF-kB-mediated inflammatory responses is a complicated phenomenon. Although, notably, Nrf2 and NF-kB interact to coordinate antioxidative and inflammatory responses and determine the destiny of innate immune cells, the mechanism by which this interconnection occurs is unknown [48].

## 6. Current FDA-Approved AD Medications

In AD, progressive destruction of neurons may reduce neurotransmitters, causing an imbalance in the levels of acetylcholine, dopamine, and serotonin, which promotes cognitive deficiencies seen in AD patients [66]. The current known treatments try to compensate for the disease’s neurotransmitter imbalance. Some of these treatments are acetylcholinesterase inhibitors (AChEIs) such as donepezil, galantamine, and rivastigmine, FDA-approved for the treatment of AD [66,67]. These drugs have an important role at synapses, increasing the acetylcholine availability, which was proven to be clinically helpful in delaying cognitive decline in AD patients [68]. The development of these drugs was based in the cholinergic hypothesis, which proposes that the progressive loss of limbic and neocortical cholinergic innervation in AD is crucial for the decline in attention, memory, learning, and other functions of the brain [68].

Another therapeutic FDA-drug approved for moderate to severe AD is the non-competitive N-methyl-D-aspartate (NMDA) receptor antagonist memantine [66,67], which binds to open NMDA receptor-operated calcium channels and blocks NMDA-mediated ion flux, which improves the harmful effects of high levels of glutamate that may lead to neuronal dysfunction [69]. 

The most common adverse effects include diarrhea, nausea, vomiting, and sometimes rapid eye movement sleep behavior disorder, which are induced by the cholinomimetic action of the AChEIs on the gastrointestinal tract. Additionally, the use of rivastigmine through a transdermal patch can cause rashes. Usually, 5 to 20% of patients present adverse effects, but they are mostly transient and mild. Bradycardia can also be induced by AChEIs, which may increase the risk of syncope. Therefore, AChEIs are not recommended for patients with severe cardiac arrhythmias, active peptic ulcers, or a history of gastrointestinal bleeding and uncontrolled seizures [70,71]. Because of this, there is a need to investigate natural compounds that could offer some brain protection and prevent or slow down the progression of the disease with fewer side effects.

## 7. The Use of Natural Compounds to Prevent Alzheimer’s Disease

Identifying novel medicines that can reduce the brain’s microglia-derived excessive OS has emerged as a key goal in the search for disease-modifying therapies to slow or stop the progression of ND in AD. Not only because of concerns about the side effects of conventional medicine but also because natural products are comparatively inexpensive and typically readily available in an ingestible form, there has been a rapid growth in interest in naturally occurring phytochemical compounds with antioxidant and neuroprotective potential over the last few decades [72]. Medicinal plants have been used as effective therapies for various illnesses and disorders for thousands of years; they are mainly considered harmless and thus chosen over conventional medicine. Furthermore, over 25% of medications used in the last 20 years are directly derived from plants, with another 25% being chemically altered natural products. This validates the study of natural products to discover chemicals that could slow or stop AD advancement. As a result, natural compounds that may alleviate OS and neuroinflammation disorders are currently being investigated [72]. Natural compounds are organic molecules with a low molecular weight (typically less than 3000 Daltons) extracted from medicinal plants and classified as “secondary metabolites. In recent years, the physicochemical parameters of natural chemicals revealed that these compounds have a wide variety of bioactive substances with great health benefits [73].

### 7.1. Cardamonin

CD is a chalcone, a type of aromatic ketone derived from the Zingiberaceae family of plants. The presence of an unsaturated ketone with two aromatic rings distinguishes this molecule. The analogs of CD identified are 2, 4,-dihydroxy-6-methoxy chalcone (DHMC) and 4, 4-dihydroxylchalcone [74]. CD is a chalcone used for thousands of years to treat digestive system problems. It possesses anti-inflammatory, antioxidant, antibacterial, anticancer, and vasorelaxant properties [75]. CD is a chalconoid found in *Alpinia katsumadai*, *Alpinia conchigera*, and *Alpinia gagnepainii*, among other species [76]. Chalcones are flavonoid aromatic enones (core structure 1,3,-diphenyl-2-propanone). These compounds get their names from the yellow coloration they provide for their respective plants and are found in the fruits, petals, leaves, bark, heartwood, and roots of plants. CD has the benefit of occurring naturally, frequently in plants already in our diet [77]. The capacity of some chalcones to limit the expression of iNOS and COX-2 and, consequently, the formation of nitric oxide and PGE2 demonstrates their anti-inflammatory properties. Furthermore, in activated macrophages, CD has been shown to decrease the release of pro-inflammatory cytokines such as TNF-α [78]. Several authors have attempted to explain CD’s anti-inflammatory activity by studying its effects on the NF-kB signaling pathway, which controls DNA transcription and is important in controlling immune responses to infection, as well as the mechanism of action on LPS-induced inflammatory gene expression [77]. CD, from *Alpinia conchigera* Griff, inhibited NF-kB activation in a quest for NF-kB inhibitors from natural sources (Zingiberaceae). The effect of CD on NF-kB activation in LPS-stimulated RAW264.7 cells and LPS-induced mortality was observed in this study. In a dose-dependent manner, this chemical significantly reduced the increased expression of the NF-kB reporter gene induced by LPS or TNF-α [79]. In RAW264.7 cells, CD administration dramatically reduced LPS-induced generation of TNF-α and NO, as well as the expression of inducible nitric-oxide synthase and COX2. CD also prevented LPS-induced degradation and phosphorylation of inhibitor kappa beta alpha, activation of Iκβ kinases, and nuclear translocation of NF-kB. CD did not inhibit Iκβ kinases directly but significantly limited activation of LPS-induced Akt. CD pre-treatment also prevented LPS-induced mortality in C57BL/6 mice, as well as a reduction in TNF-α levels in the blood. CD may be a good choice for treating NF-kB-dependent pathological conditions, such as neuroinflammation [79].

Recent research has found that CD interacts with cellular targets, such as Nrf2, to help the fight against cancers advance. CD treatment of interleukin-1-stimulated chondrocyte cells reduced the production of PGE-2 and nitric oxide and the expression of pro-inflammatory proteins, such as iNOS and COX2. CD was also found to inhibit matrix metalloproteinase activation and synthesis, reduce the activation of the NF-kB signaling pathway, and increase the amount of the antioxidant proteins heme oxygenase-1 and NADPH quinone oxidoreductase 1; all of which result in antioxidant effects [80].

### 7.2. Cardamonin as a Neuroprotective Compound

The transcription factor Nrf2 significantly impacts cellular redox and metabolic systems. Through the activation of the antioxidant defense system, reduction in inflammation, improvement of mitochondrial function, and regulation of protein homeostasis, Nrf2 activation may be a viable therapeutic strategy for neuroinflammatory illnesses. Recent research has shown that numerous plant-derived active substances can activate Nrf2 and have neuroprotective effects in various experimental paradigms, increasing the idea that Nrf2 activation could be a useful therapeutic strategy for neuroinflammatory illnesses [81]. The results of several in vitro and in vivo tests have demonstrated that natural substances can activate Nrf2 to have neuroprotective effects in the neural system. This leads us to conclude that natural/herbal compounds can potentially treat and prevent neuroinflammatory illnesses because of their capacity to activate the Nrf2 pathway. Recent research has demonstrated that specific natural compounds or ones derived from them can protect brain cells from OS. Few investigations have examined the possibilities of treating neurodegenerative diseases by activating the Keap1–Nrf2 pathway in conjunction with certain herbal plants [81].

CD’s pharmacological properties include antioxidant, antineoplastic, anti-inflammatory, anti-infectious, hypoglycemic, vasorelaxant, and autophagy induction. Recent investigations have described CD as a potent small molecule that activates Nrf2 [82]. According to the study report, Nrf2 phosphorylation and/or the oxidation/alkylation of main thiols in Keap1 may be the processes enabling CD activation of Nrf2. Other research results demonstrated that in PC12 cells treated with H_2_O_2_, CD increased phase 1 enzymes controlled by Nrf2 in a dose-dependent manner [82]. This encourages CD as a potential treatment option for preventing neurodegenerative disorders brought on by OS [81]. Since most approved therapeutics only target one gene, protein, or pathway, they are not as effective as expected because chronic diseases are frequently caused by disruptions in multiple cellular components involved in various biological processes [82]. To prevent and treat these diseases, it is essential to pursue secure, effective, and multitargeted medications.

One such substance is CD, which has been shown to influence a variety of signaling molecules, including transcription factors, cytokines, enzymes, other proteins, and genes implicated in the onset and progression of chronic disorders [74]. Due to an overabundance of polyunsaturated fatty acids, which leave the brain vulnerable to free radical damage, OS can harm the brain. Numerous NDs, including AD and PD, are characterized by excessive ROS generation, which plays a role in their pathogenesis., The brain’s redox equilibrium must be preserved to prevent NDs. An antioxidant defense system that keeps cellular redox homeostasis in check is the Nrf2–ARE pathway. Since CD is an antioxidant, it has a significant potential for preventing and treating neurological disorders [74]. One of the main aspects that influence neurodegeneration development is neuroinflammation [82]. The primary characteristics of neuroinflammation, which encourages the production of neurotoxic chemicals, is microglial activation and pro-inflammatory cytokines, causing the gradual death of brain cells. Therefore, a promising strategy to reduce neuroinflammatory symptoms is to use innovative pharmaceutical treatments to decrease the overactivation of microglia mechanisms connected to neurodegenerative disorders. Natural chemicals developed from medicinal plants have gained popularity lately and gained a great deal of attention as promising sources of new neuroprotective medications [82].

### 7.3. The Role of Cardamonin as an Antioxidant and Nrf2 Activator

There is ample proof that OS affects brain tissue in AD patients over the course of the illness. Age-related neurodegeneration and cognitive decline are thought to be significantly influenced by OS, which is defined by an imbalance in the formation of radical ROS and antioxidative defense [83].

High levels of oxidized proteins, advanced glycation end products, lipid peroxidation end products, the formation of toxic species such as peroxides, alcohols, aldehydes, free carbonyls, ketones, and cholestenone, as well as oxidative modifications in nuclear and mitochondrial DNA, are all signs of OS in AD (Figure 2). Decreases in the brain and plasma antioxidant defense mechanism are associated with age-related memory deficits [83]. The low molecular weight-reducing equivalent glutathione, which is in charge of the cell’s natural redox potential, is a crucial component of the antioxidant defense system. Its primary role is glutathione’s ability to scavenge ROS and donate electrons to them. Age-related reductions in intracellular glutathione (GSH) concentrations have been seen in various animal models, and in aging mammalian brain areas, including the hippocampus. When GSH levels drop, OS is more likely to occur because the rate of ROS creation outpaces the body’s capacity to neutralize it. An imbalance among the radical detoxification enzymes in AD is another cause of OS [83].

Activation of Nrf2 may be a functional treatment approach for neuroinflammatory illnesses, as it boosts antioxidant defenses, reduces inflammation, improves mitochondrial function, and restores protein homeostasis [81]. Many medicinal substances produced from plants have been identified to activate Nrf2 and exert neuroprotective effects in diverse animal models in recent investigations, increasing the idea that Nrf2 activation could be a viable therapeutic approach for neuroinflammatory illnesses. Natural substances produce neuroprotective effects in the neural system via activation of Nrf2, according to the results of various in vitro and in vivo studies. As a result of their ability to activate the Nrf2 pathway, herbal/natural moieties have the potential to fight and prevent neuroinflammatory illnesses [81].

The transcriptional regulation of numerous ARE-dependent antioxidant defense genes is induced when Nrf2 signaling is activated. A study by Peng et al., 2017, showed that CD, isolated from *Alpinia katsumadai*, inhibited cell death in PC12 cells generated by hydrogen peroxide (H_2_O_2_) and 6-hydroxydopamine (6-OHDA) [84]. Pre-treatment of PC12 cells with CD increased the expression of phase II antioxidant molecules controlled by Nrf2 in a dose-dependent manner. CD, on the other hand, failed to produce neuroprotection when Nrf2 expression was silenced, suggesting that this cytoprotection is mediated by Nrf2 activation [84]. It was shown that CD is a novel small chemical activator of Nrf2 in PC12 cells, implying that CD could be a promising candidate for preventing OS-related ND diseases [84].

In many ND diseases, the Nrf2 gene has been shown to be mutated. Several compounds isolated from natural plants have been shown in recent studies to cause a delay in the damage of neurons and in the progression of degeneration by inhibiting the production of free radicals and inflammation by activating Nrf2. As a result, these compounds have received a lot of attention as pharmacological treatments for ND disorders [81]. The ARE pathway has been shown to inhibit neuroinflammation with a variety of natural compounds. As a result, several researchers are focusing on the pharmacological activation of Nrf2 to treat NDs. In several investigations, activating Nrf2 in natural plants and their active components has been shown to slow the progression of ND disorders. Non-canonical processes, on the other hand, can interrupt the Keap1 and Nrf2 interaction. Figure 3 describes the activation of Nrf2 in normal and stressful situations [81,85].

During OS, free radicals trigger Nrf2 to detach from Keap1, skipping proteasomal degradation and translocating to the nucleus. There, Nfr2 binds to the ARE and starts the transcription of antioxidant enzymes including heme oxygenase-1 (HO-1), glutathione peroxidase (GPx), glutathione S-transferase (GST), superoxide dismutase (SOD), catalase (CAT), glutathione reductase (GR), NAD(P)H: quinone oxidoreductase 1 (NQO1), glutamine-cysteine ligase (GCL), and glutathione synthetase (GS). These enzymes reduce the cell’s OS and free radicals. Black arrows indicate the activation of pathways; red T-bars indicate interruption of processes [85].

Nrf2 activation mitigates various pathogenic processes involved in ND disorders by upregulating antioxidant defenses, inhibiting inflammation, improving mitochondrial function, and maintaining protein homeostasis. By modifying mitochondrial activity, Nrf2 has been implicated in the control of cellular defense mechanisms [81]. When Nrf2 is turned on, it protects mitochondrial ROS generation from poisons generated by the cells. The role of Nrf2 signaling in the production of inflammatory mediators has been elucidated, which explains the mechanism of transcriptional regulation of pro-inflammatory cytokines (TNF-α, IL-1, IL6, IL-8, and MCP-1) in microglia, macrophages, monocytes, and astrocytes after Nrf2 activation [81].

CNS illnesses are serious health problems that frequently result in disability or death. High levels of OS describe the majority of CNS illnesses. Nrf2 regulates the expression of several enzymes that have antioxidative, pro-survival, and detoxifying properties. Nrf2 forms a complex with Keap1 under normal conditions, resulting in Nrf2 inactivation via ubiquitination and destruction. However, when Keap1 is exposed to OS, Nrf2 is liberated from the protein, activated, and translocated into the nucleus. Various dietary phytochemicals have been reported to activate the Nrf2–ARE pathway [86].

Natural Nrf2–ARE activation may slow the evolution of AD by lowering OS-induced cell death [86]. Many studies have shown that activating Nrf2 impacts AD traits. The substance 2-cyano-3,12-dioxooleana-1,9-dien-28-oic acid-methyl amide (CDDO-MA), an Nrf2–ARE pathway activator, has been demonstrated to improve cognition and reduce plaque development, Aβ, and OS indicators in Tg19959 transgenic AD mice. Other Nrf2 activators, such as tBHQ and curcumin, have shown similar results. In addition to the accumulation of ROS or Aβ peptides, the administration of Nrf2 activators reduces long-term memory loss. Acetyl-L-carnitine boosted tropomyosin receptor kinase A expression and ERK phosphorylation in hypobaric hypoxia-induced dementia, resulting in Nrf2 nuclear translocation. Activating the Nrf2–ARE pathway improved memory deficits in this animal study [86].

Numerous ARE-dependent antioxidant defense genes are transcriptionally regulated when Nrf2 signaling is activated. It was demonstrated that 6-hydroxydopamine (6-OHDA) and hydrogen peroxide (H2O2)-induced cell death in PC12 cells decreased by CD treatment derived from *Alpinia katsumadai*. Pre-treatment of PC12 cells with CD increased the expression of phase II antioxidant molecules controlled by Nrf2 in a dose-dependent manner. This cytoprotection may be caused by the activation of the transcription factor Nrf2, as CD failed to offer neuroprotection when Nrf2 expression was silenced. The findings show that CD is a novel small chemical activator of Nrf2 in PC12 cells and imply that CD may be a possibility for the treatment of neurodegenerative diseases caused by OS [84].

#### Neuroprotective Effects of Nrf2 Activation

In AD, OS plays a key role in the disease’s pathogenesis and progression, with the Nrf2–Keap1–ARE pathway playing a dominant role. Nrf2/Keap1–ARE has been proven to be an upstream regulator of oxidative pathways, as well as a regulator of inflammatory and apoptotic pathways. As a result, creating multi-target drugs with superior efficacy and less side effects could help to prevent and manage AD. Natural secondary metabolites targeting Nrf2–Keap1 are currently abundant in the plant kingdom. Natural entities such as phenolic compounds, alkaloids, terpene/terpenoids, carotenoids, sulfur substances, and other assorted plant-derived compounds have shown promise. Evidence suggests that activating Nrf2–ARE and inducing antioxidant enzymes, and suppressing Keap1, could help people overcome AD [79,87].

In ND illnesses, OS appears to be an upstream driver of ND by triggering inflammation and apoptosis. Overactivated ROS/RNS have been proven to have a vital role in cell death during pathological situations. As a result, mitochondrial oxidative phosphorylation and associated pathways that are dysregulated could be important sources of generated ROS/RNS. Aβ is one of the main AD markers that plays a crucial role in activating oxidative mediators. Yet, Nrf2 is increasingly recognized as a critical upstream oxidative defense moderator. Through phosphorylation, several linked mediators can activate Nrf2 and initiate antioxidative responses. PI3K, JNK, ERK, and MAPKs are some of the main kinases that phosphorylate Nrf2 and allow it to translocate to the nucleus. During pathological situations, specific kinases are overactivated, causing Nrf2 to be degraded in both Keap1-dependent and Keap1-independent ways. Glycogen synthase kinase 3-beta (GSK-3β) has been found to degrade Nrf2 in the proteasome, whereas p38 MAPK stabilizes the Keap1/Nrf2 connection in the direction of oxidative responses. GSK-3β is essential in the progression and pathogenesis of AD. From a molecular standpoint, GSK-3β is linked to Aβ deposition and tau hyperphosphorylation, which are linked to AD pathogenesis. GSK-3β also affects OS, a prominent AD hypothesis [87].

NF-kB has also been described to co-transport with Keap1 into the nuclei trapping Nrf2. Activation of Nrf2 may regulate inflammatory pathways by activating anti-inflammatory mediators, such as IL-10, while suppressing inflammatory mediators (TNF-α, IL-6, and IL-1β). Furthermore, Nrf2 has been demonstrated to have inhibitory and stimulatory effects on apoptotic and anti-apoptotic factors. The Nrf2–ARE pathway is downregulated in hippocampus neurons during AD. As a result, increasing Nrf2–ARE could be useful for treating AD. Drugs with many targets for therapy are essential since Nrf2 is connected to numerous dysregulated pathways. Given the importance of Nrf2–ARE in avoiding AD pathogenesis, numerous types of phytochemicals have shown promise in aiming those mediators and therefore treating the disease. Phenolic chemicals, terpenes/terpenoids, alkaloids, carotenoids, and sulfur compounds are among the most important plant-derived secondary metabolites [87].

The role of Nrf2 in regulating antioxidant genes has been found to be neuroprotective in various neuronal injury and degeneration models. Increased Nrf2 activity protected cells from SH-SY5Y human neuroblastoma from oxidative injury caused by Parkinson’s neurotoxin 6-OHDA. 6-OHDA was utilized to demonstrate Nrf2 neuroprotection in in vivo and in vitro models. Another model where Nrf2 activation protects against OS is mixed neuron/astrocyte cultures from mice. In the MPTP animal model of Parkinson’s disease, Nrf2 activation reduces dopamine cell loss and striatal dopamine depletion. A transgenic AD mouse model revealed a decreased toxicity after either adenoviral Nrf2 expression or tert-butylhydroquinone stimulation of Nrf2, in addition to Nrf2 activation being neuroprotective in the previously reported PD models [88].

The increased OS and early embryonic mortality seen in combination Nrf1/Nrf2 knockout mice further demonstrate the vital relevance of Nrf2 in managing OS. Although Nrf2 knockout does not cause embryonic death, it does make these animals more susceptible to OS. Finally, Johnson and colleagues demonstrated that inducing Nrf2 in astrocytes was sufficient to rescue neurons from death caused by mutant SOD1, MPTP, or malonate-induced complex II inhibition in vivo. Given the neuroprotective benefits of Nrf2 activation, it is fair to believe that nutraceutical Nrf2 inducers, such as allicin and L-sulforaphane, could be useful in treating NDs [88].

### 7.4. Cardamonin and Neuroinflammation

Within the CNS, neuroinflammation is an aspect of the immune response that helps to eliminate pathogen-induced necrotic cells and tissues. NDs, such as AD, PD, MS, HD, and ALS, are all linked to excessive neuroinflammation. The primary hallmark of neuroinflammation is aberrant microglia activation, which promotes the production of numerous neurotoxic mediators and pro-inflammatory cytokines, resulting in progressive neuronal cell death. As a result, microglia function modification is critical for the development of safe and effective anti-inflammatory and neuroprotective medicines. Natural chemicals have attracted a lot of attention as potential sources of new therapeutic agents for neurological illnesses to treat neuroinflammation [82].

Numerous plant species have been discovered to exhibit neuroprotective properties. Natural chemicals of plant origins have recently received increased interest for their possible neuroprotective benefits in protecting cells against neuroinflammation. These small chemical compounds can disrupt protein–protein interactions, affecting biological processes that may be altered in disease conditions [82].

Natural flavonoids have been proven to have neuroprotective benefits by suppressing the release of pro-inflammatory cytokines. Flavonoids exert an anti-inflammatory impact by interfering with the production of inflammatory mediators. CD inhibited the release of pro-inflammatory mediators, such as NO, PGE-2, TNF-α, IL-1β, and IL-6, in the microglial cell line BV-2. CD also suppressed TNF-α, IL-1β, and IL-6 at both the protein and mRNA levels and inhibited the NF-kB signaling pathway by inhibiting NF-kB DNA binding activity. CD interestingly had a consistent inhibitory effect on CD14 cell surface expression [89]. These findings provide mechanistic insights into CD’s anti-inflammatory activities in BV-2 cells and, as a result, indicate a potential therapeutic application of CD for neuroinflammatory diseases [89].

### 7.5. Cardamonin and Autophagy

Autophagy is a ubiquitous catabolic cellular mechanism that maintains cellular homeostasis and quality control of the cytoplasm in response to food deprivation and environmental stimuli. Lysosomal breakdown of cellular components, including misfolded proteins or damaged organelles, is involved. Despite being predominantly a protective process for the cell, autophagy can also contribute to cell death. Autophagy has a role in both the innate and adaptive immune systems, and its failure is linked to inflammation, infection, neurodegeneration, and cancer [90]. Traditional plant medicines with new therapeutic uses have been increasingly popular in recent years, not just in Western but also in Chinese herbal medicine (CHM). Pharmacological studies have shown that active components or fractions from medicinal plants known as Chaihu (*Radix bupleuri*), Hu Zhang (*Rhizoma polygoni cuspidati*), Donglingcao (*Rabdosia rubesens*), Hou po (*Cortex magnolia officinalis*), and Chuan xiong (*Rhizoma chuanxiong*) modulate cancers, ND, and offer insight into the possibility of new CHM decoction applications and formulations through regulating autophagy [91].

The two major proteins that assemble in AD, tau and Aβ, are autophagy substrates. Tau levels are reduced when autophagy is induced. Conditional deletion of the autophagy protein ATG7 in the forebrains of mice resulted in phospho-tau buildup in a pattern similar to that of a pre-tangle state. While tau deletion does not prevent inclusion formation, it does help these mice recover from neurodegeneration. Autophagy has been hypothesized to play a crucial function in the metabolism of Aβ, but it may also play a role in its production. Both APP and PS1 have been found in the autophagic vesicles that accumulate in AD neurons. Autophagy has also been linked to Aβ secretion, as crossing APP transgenic mice with animals lacking ATG7 in forebrain neurons resulted in reduced Aβ extracellular secretion and plaque formation. As a result of both a decrease in clearance and a decrease in protein release, the loss of autophagy may result in increased intracellular Aβ. The involvement of autophagy in AD is overly complex, and this could be due to the fact that autophagy has distinct effects at different stages of the disease, as well as the potential that autophagy affects various stages of the amyloid life cycle [92].

Inflammation, autophagy, and AD appear to be linked processes, according to current knowledge. Francois et al. showed crosstalk between them in their in vitro studies, where Aβ42 affected the way several autophagy proteins (p62, p70S6K) were expressed and activated [93]. They also found that inflammatory processes and autophagy were associated inside brain cells, as IL-1β-induced acute inflammation stimulated autophagy in microglia cultured in tri- or mono-cultures. As a result, it is possible that IL-1β plays a role in the etiology of AD by inducing both neuroinflammation and autophagy [94].

Immune signals may trigger autophagy during the course of AD. Indeed, following stress-induced hypertension, neuroinflammation has been demonstrated to impact autophagy. Adult mice with mutations in the *App* and *Psen1* genes had greater brain levels of inflammatory mediators (including Il-1β) and an accumulation of autophagic vesicles within dystrophic neurons in the cortex and hippocampus. Furthermore, inflammatory mediator levels were linked to the expression of critical regulators of autophagy including mTOR and Becn1. In vitro, however, it was reported that inhibiting autophagy increases microglia activity, including releasing cytokines, such as Il-1β, and creating harmful ROS. These findings show that AD and neuroinflammation feed autophagy, whereas autophagy reduces brain inflammation. As a result of its interaction with the immune system, an increase in autophagy may serve a protective function during the progression of AD [94].

Many studies have shown that flavonoids’ active components can alter autophagy in various illnesses. The plant flavonoid silibinin, isolated from *Silybum marianum*, improved Aβ1-42-induced depression in rats and relieved neuronal damage in the hippocampus by suppressing autophagy. Wogonin, a flavonoid derived from *Scutellaria baicalensis*, improves Aβ clearance and lowers Aβ accumulation in cortical astrocytes by regulating autophagy. Furthermore, hesperetin and its glycoside hesperidin have been linked to neuronal protection by inhibiting Aβ-mediated autophagy. A range of *Panax ginseng*-derived medicines have also been found to promote neuroprotection and improve cognitive performance in dementia patients. In a mouse model of AD, Rg2 ginseng-induced autophagy accelerated the clearance of aggregated proteins, reduced the accumulation of cerebral Aβ, and improved cognitive functioning via autophagy [95].

The rapamycin’s mammalian target (mTOR) incorporates energy level and is crucial for autophagic activity. Due to autophagy’s dual function, mTOR depends on whether inhibition is positive or negative, thus a cell may die or survive. This protein functions as a cell death inducer or a means of survival. There is strong evidence that mTOR inhibition triggers the start of autophagy even when there are enough calories and growth hormones present. Flavonoid chemicals may reduce mTOR’s interaction with its subsequently lower phosphorylation of component S6K1. This offers fresh approaches for making mTOR inhibitors [96].

Additionally, CD has been shown to impede nuclear translocation and have antiproliferative effects in various cancer cells. CD’s anti-tumor mechanism is still not fully understood. However, it appears to include NF-kB and mitogen-activated protein kinase. According to certain studies, CD suppresses mTOR to prevent Lewis lung carcinoma metastasis and the proliferation of A549 cells. CD suppresses mTOR without the use of FK506 binding protein, in contrast to the traditional mTOR inhibitor (rapamycin) 12 kDa (FKBP12). Therefore, it is necessary to describe the precise mechanism through which CD inhibits mTOR. However, it is important to understand how CD affects the autophagy of ovarian cancer cells. The impact of CD on SKOV3 cells’ proliferative, autophagic, and apoptotic processes was investigated. Researchers identified the protein levels of the mTOR complex components connected to the activation of the mTOR signal pathway to understanding further the underlying mechanism of CD on mTOR inhibition [96].

In tests, CD’s effects on cell proliferation, cell cycle distribution, and autophagy stimulation in cultures of the HCT116 cell line were examined. CD was found to reduce cell growth in HCT116 cells, induce cell cycle arrest in the G2/M phase, and increase autophagy [24]. Researchers discovered that the tumor protein p53 controls the autophagic and antiproliferative effects of CD. They also discovered that p53 partially controlled the enhanced CD-induced activation of c-Jun N-terminal kinase (JNK), which was necessary for the drug’s autophagic and antiproliferative actions on HCT116 cells. These results show that colorectal cancer patients may benefit from treatment with CD or other anticancer drugs that promote p53/JNK-dependent stimulation of autophagy [24].

### 7.6. Cardamonin and Microglia

Microglia have a peculiar, ramified shape in healthy CNS tissue, with spherical soma and multiple branching processes. Although long thought to be in a “resting” state, new data suggests that ramified microglia play important physiologic roles during brain and spinal cord development, including neuronal fate determination, migration, axonal growth, and synaptic remodeling [82]. Microglia play an important role in CNS development due to their phagocytosis of cellular debris, the release of a range of cell signaling molecules such as neurotrophins and extracellular matrix components, and direct interaction with neurons. Microglia also have the mechanisms and properties to function as sensors in the mature CNS for disruptions in normal homeostasis. Various pathologic conditions, including altered neuronal function, infection, injury, ischemia, and inflammation, cause microglia to become quickly activated. When microglia are activated, their shape changes to that of an ameboid, making it easier for them to migrate to the injury site. In addition to phagocytosis, antigen processing and presentation, and the generation of both cytotoxic and neurotrophic factors, microglia responses to CNS disease result in the activation of many immunological functions [82].

Microglia are activated during neuroinflammation, which releases neurotoxic chemicals and pro-inflammatory cytokines and gradual neuronal cell death. Thus, employing innovative pharmacological treatments to reduce microglia overactivation is an appealing topic for alleviating the neuroinflammatory processes associated with dementia. Natural chemicals derived from medicinal plants, such as flavonoids, glycosides, phenolics, terpenoids, quinones, alkaloids, lignans, coumarins, chalcone, stilbene, and other natural substances, are effective in preventing microglia activation (biphenyl, phenylpropanoid, oxy carotenoid). They can inhibit the expression of neurotoxic mediators (NO, PGE2, iNOS, COX-2) and pro-inflammatory cytokines (IL-6, TNF-α, IL-1β), as well as down-regulate inflammatory indicators and prevent brain injury. Experimental research shows that they have anti-neuroinflammatory actions via regulating important signaling pathways (NF-kB, MAPKs, Nrf2/HO-1, PI3K/Akt, JAK/STAT) [88]. 

The chalcone CD isolated from *Alpinia rafflesiana* reduced inflammatory responses in BV-2 microglia activated with interferon (IFN-γ)/LPS via the NF-kB signaling pathway. NF-kB has been demonstrated to suppress the transcription of genes controlled by Nrf2. In neuroinflammation, there appears to be a crosstalk between these transcription factors when taken together [82].

The pathophysiology of various inflammation-related disorders has been linked to abnormal activation of the NLRP3 inflammasome, and pharmacological compounds that target the NLRP3 inflammasome could be useful in identifying possible therapeutic approaches. CD from *Alpinia katsumadai* has shown to have excellent anti-inflammatory properties. CD is a broad-spectrum NLRP3 inflammasome inhibitor that is activated by a variety of triggers. Furthermore, the effect of CD on inflammasome activation is limited to the NLRP3 inflammasome, not the NLRC4 or AIM2 inflammasomes. Importantly, CD helps mice survive LPS-induced lethal endotoxic shock, which has been proven to be NLRP3-dependent [97].

### 7.7. Cardamonin and MicroRNA

MicroRNAs are non-coding RNAs that adversely bind to the 3′ untranslated regions (UTR) of target messenger RNAs to influence gene expression at the post-transcriptional stage. MicroRNAs are important and necessary in pulmonary developmental and pathological processes, according to new findings. MicroRNA-377-3p (miR-377-3p) activated protective autophagy and decreased LPS-induced inflammation and lung damage. According to Chen et al., miR-199a-3p plays a protective function against LPS-induced acute lung injury (ALI), demonstrating that miR-199a-3p down-regulation exacerbated intrapulmonary inflammation and pathological damage [98]. miR-23a-5p regulates cell proliferation, differentiation, senescence, survival, and oncogenesis, among other biological processes [99].

Inflammation and OS are also regulated by miR-23a-5p. After LPS stimulation, Liu et al. discovered increased miR-23a-5p levels in serum, lung tissues, and macrophages and recommended miR-23a-5p as a possible biomarker for sepsis-induced acute respiratory distress syndrome. Its therapeutic role in inflammation, OS, and ALI, however, is still unknown [99].

Although the pathophysiology of vascular dementia (VaD) is still unknown, non-coding RNAs such as microRNAs (miRNAs), long non-coding RNAs, and circular RNAs have been implicated in vascular dysfunction, impairment of cognition, neuroinflammatory reaction, disruption of the blood–brain barrier, and loss of synapses. miRNAs are abundant in the brain and are involved in the post-transcriptional regulation of a broad spectrum of proteins by degrading or inhibiting their target mRNAs. In a rat model of VaD, a decrease in miR-126 expression in endothelial cells showed the pathophysiological process mediating impairment of cognition in the rats, which was followed by a reduced flow of cerebral blood and vessel patency, causing activation of astrocytes and microglia and a decrease in synaptic plasticity [100]. As cognition deficits in VaD were treated with acupuncture, the miR-93-mediated Toll-like receptor signaling pathway played an important role. *Ginkgo biloba* L., *Salvia miltiorrhiza* Bunge, *Panax ginseng* C.A. Mey. and *Dracocephalum moldavica* L. (*D. moldavica*) are among the almost one hundred traditional Chinese remedies used to treat VaD. Many of these herbal medications, when used to treat vascular cognitive decline, have roles and mechanisms in the regulation of abnormal miRNAs. For example, EGb761, an extract from *Ginkgo biloba* L., protected brain microvascular endothelial cells from ischemia/reperfusion injury via the lncRNA Rmst/miR-150 axis, miR-155-5p, miR-144, and miR-216a are all inhibited by ginsenoside Rg1 and Rg2 isolated from *Panax ginseng* C.A. Mey [100].

A traditional Chinese Uygur medicine, *D. moldavica*, was used to cure coronary heart disease, angina, and atherosclerosis and soothe nerves [100]. Modern neuropharmacological studies have confirmed that *D. moldavica* has favorable effects on the CNS, as evidenced by the induction of extended pentobarbital-induced sedation in mice, neuroprotection against cerebral ischemia/reperfusion injury in rats, and cognitive improvement in mice with scopolamine-induced deficits. The total flavonoids from *D. moldavica* represent a standardized extract and operate as a unique treatment for VaD, with the help of a number of experts [100].

The ability of quercetin to regulate miRNA expression is the primary mechanism through which it has a substantial impact on human health. Quercetin’s protective effect is aided by this intriguing method of action. Inflammation, differentiation, proliferation, apoptosis, immunological response, and neurodegeneration all have miRNAs that regulate them. miRNAs prevent target mRNAs from being translated or degraded. It has been discovered that in neurodegenerative diseases, such as AD, the control of miRNA is disrupted [101].

One miRNA can control many genes at the same time, and multiple miRNAs can modulate the same gene at the same time. This intriguing regulation of miRNAs encourages the creation of new perspectives in the study of complicated disorders, such as AD. During the various phases of AD, numerous cells are involved in the disease with abnormal expression of miRNAs [101]. As a result, the levels of miR-26a and miR125b are increased, while the levels of miR-138, miR-132, miR219, and miRNA-15 are decreased. Indeed, miR-132 and miR-219 directly target tau mRNA expression and suppress tau production; elevation of miR-138 and miR125b, on the other hand, increases tau phosphorylation via GSK-3, ERK1/2, and CDK5 activation, respectively. miRNA-132 can protect against tauopathies [101].

In vitro studies with quercetin showed that it strongly protected PC-12 neuronal cells, which are extensively employed in neurological disease research, against hydrogen peroxide-induced mortality [101]. ROS are intimately associated with neurodegenerative diseases and function as monitors and effectors of miRNA and its target proteins. 135 miRNAs were discovered in brain cells in response to OS, and a reduced number of miRNAs was identified as a crucial factor in the antioxidant response caused by quercetin [101]. Quercetin pre-treatment of neuronal cells prevented OS-induced changes in the expression of 14 miRNAs (novel miR-2218, novel miR-724, novel miR-645, novel miR-2117, novel miR-1795, novel miR-2502, novel miR-291, novel miR-1502, novel miR-2766, novel miR-1387, novel miR-345, novel miR-298, novel miR-521, and novel miR-97) induced by OS [101]. It is worth noting that all of these miRNAs associated with quercetin’s antioxidant properties were only recently discovered. As a result, more research is needed to understand their roles in neurodegeneration [101].

Currently, the third most common cause of cancer-related mortality is colorectal cancer. The use of dietary intervention tactics to fend against chronic illnesses, such as cancer, has garnered a lot of interest. CD is a nutraceutical produced from spices, and in this research, its therapeutic potential was assessed in a mouse model of colorectal cancer generated by the chemical azoxymethane (AOM). Four groups of mice were created, and three of those groups received six weekly injections of AOM. The other groups received either a vehicle or CD treatment beginning on the same day or 16 weeks following the initial AOM injection, with one group acting as an untreated control [102]. Treatment with CD reduced the number of Ki-67- and β-catenin-positive cells, as well as the occurrence and spread of tumors. CD therapy also resulted in the abrogation of NF-kB signaling activity. A global microRNA profiling of the colon sample was conducted to clarify the mode of action. A computational investigation showed that the groups’ miRNA expression was different from one another. The growth, cell cycle arrest, and apoptosis in human colorectal cancer cell lines were all decreased by CD, according to research findings, when they expanded the research to include human colorectal cancer [102].

## 8. Conclusions

There is currently a lot of interest in adopting dietary intervention approaches to prevent or delaying neurodegenerative diseases, such as AD. CD’s activation of the Keap1–Nrf2–ARE pathway appears to be the primary mechanism by which it produces antioxidant, anti-inflammatory, neuroprotective, and cytoprotective effects. Previous articles have described that CD modulates a variety of signaling molecules involved in the development and progression of chronic diseases, including transcription factors (Nrf2, NF-kB, and STAT3), cytokines (TNF-α, IL-1, and IL-6), enzymes (COX-2, MMP-9, and ALDH1), and other proteins and genes (Bcl-2, XIAP, and cyclin D1). CD has been reported to have multiple therapeutic properties, including anti-cancer, anti-inflammatory, antioxidant, antiviral antibiotic, antifungal, and antiallergic activities. Due to its antioxidant, anti-inflammatory, and neuroprotective characteristics, CD has been demonstrated to limit the direct harmful effects of oxidants and change the underlying inflammatory processes that play a significant part in the development of neurodegenerative diseases, such as AD. As a result of the documented pharmacological properties, CD appears to be a promising compound with the potential to be used in AD-preventive therapy (Figure 4).

## Figures and Tables

**Figure 1 brainsci-13-00145-f001:**
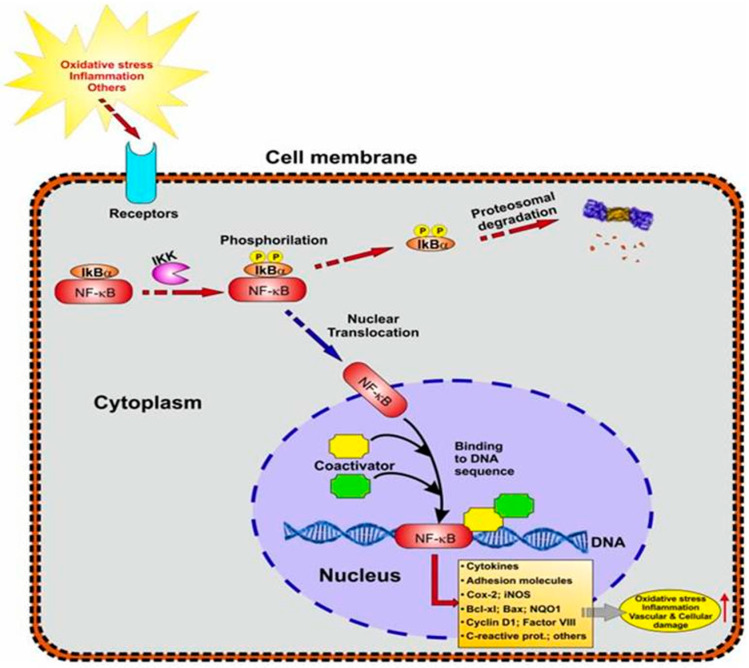
NF-kB regulation under cellular stress conditions: NF-kB is a transcription factor that modulates immune function, inflammation, cellular growth, and apoptosis. OS induces activation of Iκβ kinase (IKK), leading to the phosphorylation of NF-kB inhibitor, resulting in polyubiquitination-mediated proteasomal degradation of Iκβ, releasing NF-kB, which migrates into the nucleus, binds to its corresponding DNA responsive elements and induces the transcription of pro-inflammatory mediators [48].

**Figure 2 brainsci-13-00145-f002:**
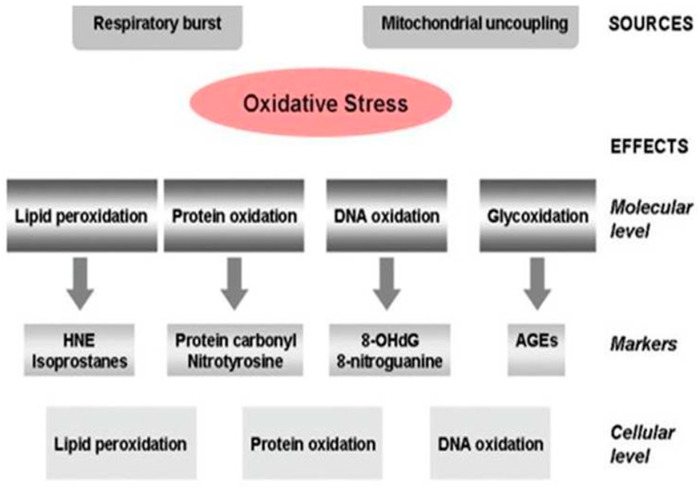
Oxidative stress sources and effects at the molecular and cellular levels, as well as their corresponding markers [83].

**Figure 3 brainsci-13-00145-f003:**
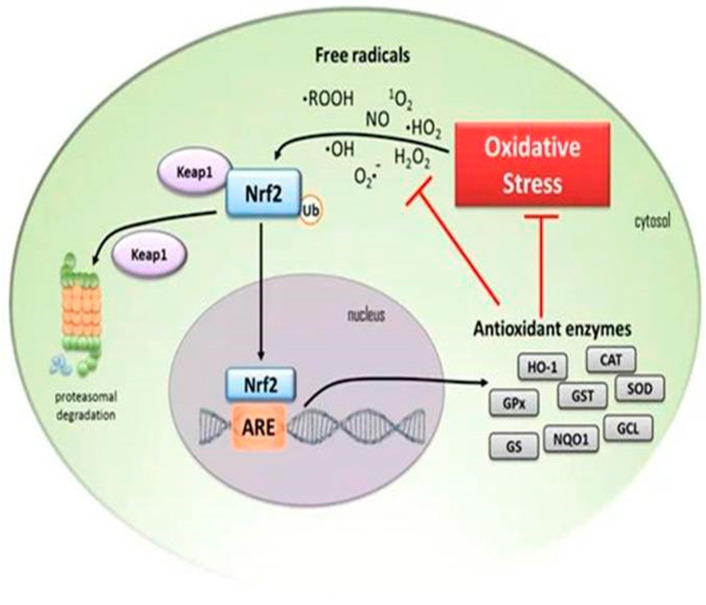
The activation of Nrf2-inducing antioxidant genes. Under a homeostatic state, cytosolic Nrf2 is maintained at reduced levels by proteasomal degradation induced by the Keap1 protein complex.

**Figure 4 brainsci-13-00145-f004:**
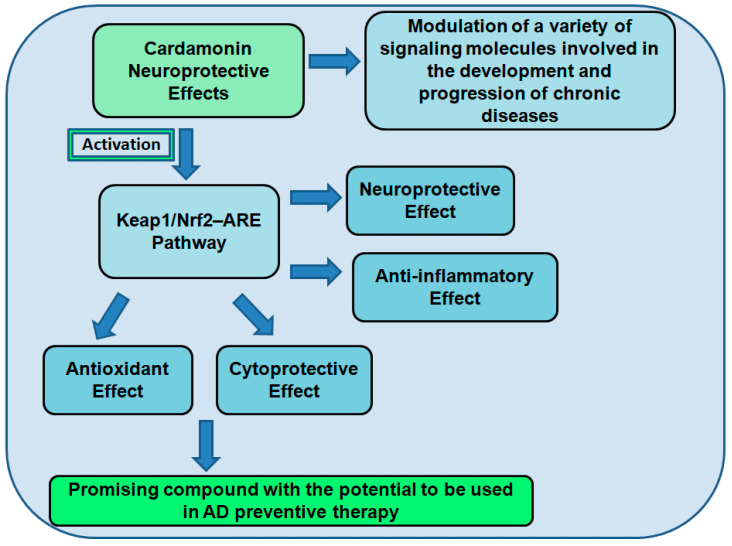
Summary of CD’s neuroprotective effects. Studies show that CD modulates signaling molecules including transcription factors, cytokines, protein and genes. CD also activates Nrf2 signaling inducing antioxidant, anti-inflammatory, and cytoprotective effects, indicating its potential as a cytoprotective therapeutic compound against AD.

## Data Availability

Not applicable.

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
