# Peer review of "The Neuroprotective Effects and Therapeutic Potential of the Chalcone Cardamonin for Alzheimer’s Disease"

_brainsci, 2023, doi:10.3390/brainsci13010145_

Round 1
Reviewer 1 Report
This is a very interesting review focusing on the neuroprotective effects of CARDAMONIN for the treatment of neurodegenerative diseases. The paper is well-written and of interest for the journal. However, several changes are recommended before considering it for publication.
This is a narrative review mainly focusing on Alzheimer's Disease and Parkinson Disease. I recommend to add a Methods section explaining how have the authors selected and screened the papers. Although it is a narrative review, the methods are important. Which were the inclusion and exclusion criteria?
Section 2 is mainly focusing on neurodegenerative diseases, Alzheimer's Disease, section 3 about pathogenesis and section 4 about Oxidative Stress.
I think it would be better to group these three sections into 1 to better explain the Alzheimer's disease into the context. I consider that a specific section about Biomarkers would be helpful. Biomarkers for diagnosis and treatment would be interesting. Section 6 is mainly focused on neuroinflammation. This could be also grouped under "Biomarkers".
In section 7, the authors are explaining the main compound to prevent Alzheimer's Disease. Before explaining the different strategies to prevent AZ, I recommend to explain other biological strategies, others than those derived from plants.
Line 853. Please adapt the reference from François to the journal style. The year is not necessary.
In the body of the manuscript there is a lack of information about the effects of cardamonin on Parkinson's disease. If the paper is mainly focused on Alzheimer's disease, the title and the objectives should be restricted to AD.
At the end of the conclusions section the authors have included some information from the template (This section is not mandatory...).
Author Response
Dear Editor:
We are pleased to resubmit the revised version of the Manuscript ID: brainsci-2122210, title: "The Neuroprotective Effects and Therapeutic Potentials of the Chalcone Cardamonin for Neurodegenerative Diseases." We have changed the manuscript and made the requested modifications. We appreciate the reviewers' constructive criticisms and comments and have addressed each of their concerns as outlined below.
Reviewer 1:
1) This is a very interesting review focusing on the neuroprotective effects of CARDAMONIN for the treatment of neurodegenerative diseases. The paper is well-written and of interest to the journal. However, several changes are recommended before considering it for publication. This is a narrative review mainly focusing on Alzheimer's Disease and Parkinson's Disease. I recommend adding a Methods section explaining how the authors have selected and screened the papers. Although it is a narrative review, the methods are important. Which were the inclusion and exclusion criteria?
Response: As recommended by the reviewer, we included the methods used to select the papers cited in this review (Section 2 - Methods for Screening Cited Papers).
2) Section 2 mainly focuses on neurodegenerative diseases, Alzheimer's Disease, section 3 about pathogenesis, and section 4 about Oxidative Stress. I think it would be better to group these three sections into 1 better to explain Alzheimer's Disease in the context.
Response: As suggested by the reviewer, the 3 sections were combined into section 3.1 Alzheimer's Disease Pathogenesis, Risk Factors, and Oxidative Stress.
3) I consider that a specific section about Biomarkers would be helpful. Biomarkers for diagnosis and treatment would be interesting. Section 6 is mainly focused on neuroinflammation. This could also be grouped under "Biomarkers."
Response: Following the suggestion of the reviewer, a new section about Biomarkers was added to the review (Section 5. Biomarkers for Diagnosis and Treatment of Alzheimer's Disease).
4) In section 7, the authors explain the main compound to prevent Alzheimer's Disease. Before explaining the different strategies to prevent AZ, I recommend explaining other biological strategies other than those derived from plants.
Response: Following the reviewer's suggestion, a new section was added to explain the current drugs in use to treat AD and the side effects caused by these treatments (Section 6. Current FDA-approved AD medications).
5) Line 853. Please adapt the reference from François to the journal style. The year is not necessary.
Response: As requested, the reference was updated.
6) In the body of the manuscript, there is a lack of information about the effects of cardamonin on Parkinson's Disease. If the paper is mainly focused on Alzheimer's Disease, the title and the objectives should be restricted to AD.
Response: We agreed with the reviewer. The title, second and last lines of the abstract of the review were updated. The objective already states that the review describes CD properties that can lead to the prevention of ND, particularly AD.
7) At the end of the conclusions section, the authors have included some information from the template (This section is not mandatory...).
Response: We appreciate the reviewer's observation. Information from the template was removed.
Dear Editor:
We are pleased to resubmit the revised version of the Manuscript ID: brainsci-2122210, title: "The Neuroprotective Effects and Therapeutic Potentials of the Chalcone Cardamonin for Neurodegenerative Diseases." We have changed the manuscript and made the requested modifications. We appreciate the reviewers' constructive criticisms and comments and have addressed each of their concerns as outlined below.
Reviewer 1:
1) This is a very interesting review focusing on the neuroprotective effects of CARDAMONIN for the treatment of neurodegenerative diseases. The paper is well-written and of interest to the journal. However, several changes are recommended before considering it for publication. This is a narrative review mainly focusing on Alzheimer's Disease and Parkinson's Disease. I recommend adding a Methods section explaining how the authors have selected and screened the papers. Although it is a narrative review, the methods are important. Which were the inclusion and exclusion criteria?
Response: As recommended by the reviewer, we included the methods used to select the papers cited in this review (Section 2 - Methods for Screening Cited Papers).
2) Section 2 mainly focuses on neurodegenerative diseases, Alzheimer's Disease, section 3 about pathogenesis, and section 4 about Oxidative Stress. I think it would be better to group these three sections into 1 better to explain Alzheimer's Disease in the context.
Response: As suggested by the reviewer, the 3 sections were combined into section 3.1 Alzheimer's Disease Pathogenesis, Risk Factors, and Oxidative Stress.
3) I consider that a specific section about Biomarkers would be helpful. Biomarkers for diagnosis and treatment would be interesting. Section 6 is mainly focused on neuroinflammation. This could also be grouped under "Biomarkers."
Response: Following the suggestion of the reviewer, a new section about Biomarkers was added to the review (Section 5. Biomarkers for Diagnosis and Treatment of Alzheimer's Disease).
4) In section 7, the authors explain the main compound to prevent Alzheimer's Disease. Before explaining the different strategies to prevent AZ, I recommend explaining other biological strategies other than those derived from plants.
Response: Following the reviewer's suggestion, a new section was added to explain the current drugs in use to treat AD and the side effects caused by these treatments (Section 6. Current FDA-approved AD medications).
5) Line 853. Please adapt the reference from François to the journal style. The year is not necessary.
Response: As requested, the reference was updated.
6) In the body of the manuscript, there is a lack of information about the effects of cardamonin on Parkinson's Disease. If the paper is mainly focused on Alzheimer's Disease, the title and the objectives should be restricted to AD.
Response: We agreed with the reviewer. The title, second and last lines of the abstract of the review were updated. The objective already states that the review describes CD properties that can lead to the prevention of ND, particularly AD.
7) At the end of the conclusions section, the authors have included some information from the template (This section is not mandatory...).
Response: We appreciate the reviewer's observation. Information from the template was removed.
Dear Editor:
We are pleased to resubmit the revised version of the Manuscript ID: brainsci-2122210, title: "The Neuroprotective Effects and Therapeutic Potentials of the Chalcone Cardamonin for Neurodegenerative Diseases." We have changed the manuscript and made the requested modifications. We appreciate the reviewers' constructive criticisms and comments and have addressed each of their concerns as outlined below.
Reviewer 1:
1) This is a very interesting review focusing on the neuroprotective effects of CARDAMONIN for the treatment of neurodegenerative diseases. The paper is well-written and of interest to the journal. However, several changes are recommended before considering it for publication. This is a narrative review mainly focusing on Alzheimer's Disease and Parkinson's Disease. I recommend adding a Methods section explaining how the authors have selected and screened the papers. Although it is a narrative review, the methods are important. Which were the inclusion and exclusion criteria?
Response: As recommended by the reviewer, we included the methods used to select the papers cited in this review (Section 2 - Methods for Screening Cited Papers).
2) Section 2 mainly focuses on neurodegenerative diseases, Alzheimer's Disease, section 3 about pathogenesis, and section 4 about Oxidative Stress. I think it would be better to group these three sections into 1 better to explain Alzheimer's Disease in the context.
Response: As suggested by the reviewer, the 3 sections were combined into section 3.1 Alzheimer's Disease Pathogenesis, Risk Factors, and Oxidative Stress.
3) I consider that a specific section about Biomarkers would be helpful. Biomarkers for diagnosis and treatment would be interesting. Section 6 is mainly focused on neuroinflammation. This could also be grouped under "Biomarkers."
Response: Following the suggestion of the reviewer, a new section about Biomarkers was added to the review (Section 5. Biomarkers for Diagnosis and Treatment of Alzheimer's Disease).
4) In section 7, the authors explain the main compound to prevent Alzheimer's Disease. Before explaining the different strategies to prevent AZ, I recommend explaining other biological strategies other than those derived from plants.
Response: Following the reviewer's suggestion, a new section was added to explain the current drugs in use to treat AD and the side effects caused by these treatments (Section 6. Current FDA-approved AD medications).
5) Line 853. Please adapt the reference from François to the journal style. The year is not necessary.
Response: As requested, the reference was updated.
6) In the body of the manuscript, there is a lack of information about the effects of cardamonin on Parkinson's Disease. If the paper is mainly focused on Alzheimer's Disease, the title and the objectives should be restricted to AD.
Response: We agreed with the reviewer. The title, second and last lines of the abstract of the review were updated. The objective already states that the review describes CD properties that can lead to the prevention of ND, particularly AD.
7) At the end of the conclusions section, the authors have included some information from the template (This section is not mandatory...).
Response: We appreciate the reviewer's observation. Information from the template was removed.
Dear Editor:
We are pleased to resubmit the revised version of the Manuscript ID: brainsci-2122210, title: "The Neuroprotective Effects and Therapeutic Potentials of the Chalcone Cardamonin for Neurodegenerative Diseases." We have changed the manuscript and made the requested modifications. We appreciate the reviewers' constructive criticisms and comments and have addressed each of their concerns as outlined below.
Reviewer 1:
1) This is a very interesting review focusing on the neuroprotective effects of CARDAMONIN for the treatment of neurodegenerative diseases. The paper is well-written and of interest to the journal. However, several changes are recommended before considering it for publication. This is a narrative review mainly focusing on Alzheimer's Disease and Parkinson's Disease. I recommend adding a Methods section explaining how the authors have selected and screened the papers. Although it is a narrative review, the methods are important. Which were the inclusion and exclusion criteria?
Response: As recommended by the reviewer, we included the methods used to select the papers cited in this review (Section 2 - Methods for Screening Cited Papers).
2) Section 2 mainly focuses on neurodegenerative diseases, Alzheimer's Disease, section 3 about pathogenesis, and section 4 about Oxidative Stress. I think it would be better to group these three sections into 1 better to explain Alzheimer's Disease in the context.
Response: As suggested by the reviewer, the 3 sections were combined into section 3.1 Alzheimer's Disease Pathogenesis, Risk Factors, and Oxidative Stress.
3) I consider that a specific section about Biomarkers would be helpful. Biomarkers for diagnosis and treatment would be interesting. Section 6 is mainly focused on neuroinflammation. This could also be grouped under "Biomarkers."
Response: Following the suggestion of the reviewer, a new section about Biomarkers was added to the review (Section 5. Biomarkers for Diagnosis and Treatment of Alzheimer's Disease).
4) In section 7, the authors explain the main compound to prevent Alzheimer's Disease. Before explaining the different strategies to prevent AZ, I recommend explaining other biological strategies other than those derived from plants.
Response: Following the reviewer's suggestion, a new section was added to explain the current drugs in use to treat AD and the side effects caused by these treatments (Section 6. Current FDA-approved AD medications).
5) Line 853. Please adapt the reference from François to the journal style. The year is not necessary.
Response: As requested, the reference was updated.
6) In the body of the manuscript, there is a lack of information about the effects of cardamonin on Parkinson's Disease. If the paper is mainly focused on Alzheimer's Disease, the title and the objectives should be restricted to AD.
Response: We agreed with the reviewer. The title, second and last lines of the abstract of the review were updated. The objective already states that the review describes CD properties that can lead to the prevention of ND, particularly AD.
7) At the end of the conclusions section, the authors have included some information from the template (This section is not mandatory...).
Response: We appreciate the reviewer's observation. Information from the template was removed.
Dear Editor:
We are pleased to resubmit the revised version of the Manuscript ID: brainsci-2122210, title: "The Neuroprotective Effects and Therapeutic Potentials of the Chalcone Cardamonin for Neurodegenerative Diseases." We have changed the manuscript and made the requested modifications. We appreciate the reviewers' constructive criticisms and comments and have addressed each of their concerns as outlined below.
Reviewer 1:
1) This is a very interesting review focusing on the neuroprotective effects of CARDAMONIN for the treatment of neurodegenerative diseases. The paper is well-written and of interest to the journal. However, several changes are recommended before considering it for publication. This is a narrative review mainly focusing on Alzheimer's Disease and Parkinson's Disease. I recommend adding a Methods section explaining how the authors have selected and screened the papers. Although it is a narrative review, the methods are important. Which were the inclusion and exclusion criteria?
Response: As recommended by the reviewer, we included the methods used to select the papers cited in this review (Section 2 - Methods for Screening Cited Papers).
2) Section 2 mainly focuses on neurodegenerative diseases, Alzheimer's Disease, section 3 about pathogenesis, and section 4 about Oxidative Stress. I think it would be better to group these three sections into 1 better to explain Alzheimer's Disease in the context.
Response: As suggested by the reviewer, the 3 sections were combined into section 3.1 Alzheimer's Disease Pathogenesis, Risk Factors, and Oxidative Stress.
3) I consider that a specific section about Biomarkers would be helpful. Biomarkers for diagnosis and treatment would be interesting. Section 6 is mainly focused on neuroinflammation. This could also be grouped under "Biomarkers."
Response: Following the suggestion of the reviewer, a new section about Biomarkers was added to the review (Section 5. Biomarkers for Diagnosis and Treatment of Alzheimer's Disease).
4) In section 7, the authors explain the main compound to prevent Alzheimer's Disease. Before explaining the different strategies to prevent AZ, I recommend explaining other biological strategies other than those derived from plants.
Response: Following the reviewer's suggestion, a new section was added to explain the current drugs in use to treat AD and the side effects caused by these treatments (Section 6. Current FDA-approved AD medications).
5) Line 853. Please adapt the reference from François to the journal style. The year is not necessary.
Response: As requested, the reference was updated.
6) In the body of the manuscript, there is a lack of information about the effects of cardamonin on Parkinson's Disease. If the paper is mainly focused on Alzheimer's Disease, the title and the objectives should be restricted to AD.
Response: We agreed with the reviewer. The title, second and last lines of the abstract of the review were updated. The objective already states that the review describes CD properties that can lead to the prevention of ND, particularly AD.
7) At the end of the conclusions section, the authors have included some information from the template (This section is not mandatory...).
Response: We appreciate the reviewer's observation. Information from the template was removed.
Dear Editor:
We are pleased to resubmit the revised version of the Manuscript ID: brainsci-2122210, title: "The Neuroprotective Effects and Therapeutic Potentials of the Chalcone Cardamonin for Neurodegenerative Diseases." We have changed the manuscript and made the requested modifications. We appreciate the reviewers' constructive criticisms and comments and have addressed each of their concerns as outlined below.
Reviewer 1:
1) This is a very interesting review focusing on the neuroprotective effects of CARDAMONIN for the treatment of neurodegenerative diseases. The paper is well-written and of interest to the journal. However, several changes are recommended before considering it for publication. This is a narrative review mainly focusing on Alzheimer's Disease and Parkinson's Disease. I recommend adding a Methods section explaining how the authors have selected and screened the papers. Although it is a narrative review, the methods are important. Which were the inclusion and exclusion criteria?
Response: As recommended by the reviewer, we included the methods used to select the papers cited in this review (Section 2 - Methods for Screening Cited Papers).
2) Section 2 mainly focuses on neurodegenerative diseases, Alzheimer's Disease, section 3 about pathogenesis, and section 4 about Oxidative Stress. I think it would be better to group these three sections into 1 better to explain Alzheimer's Disease in the context.
Response: As suggested by the reviewer, the 3 sections were combined into section 3.1 Alzheimer's Disease Pathogenesis, Risk Factors, and Oxidative Stress.
3) I consider that a specific section about Biomarkers would be helpful. Biomarkers for diagnosis and treatment would be interesting. Section 6 is mainly focused on neuroinflammation. This could also be grouped under "Biomarkers."
Response: Following the suggestion of the reviewer, a new section about Biomarkers was added to the review (Section 5. Biomarkers for Diagnosis and Treatment of Alzheimer's Disease).
4) In section 7, the authors explain the main compound to prevent Alzheimer's Disease. Before explaining the different strategies to prevent AZ, I recommend explaining other biological strategies other than those derived from plants.
Response: Following the reviewer's suggestion, a new section was added to explain the current drugs in use to treat AD and the side effects caused by these treatments (Section 6. Current FDA-approved AD medications).
5) Line 853. Please adapt the reference from François to the journal style. The year is not necessary.
Response: As requested, the reference was updated.
6) In the body of the manuscript, there is a lack of information about the effects of cardamonin on Parkinson's Disease. If the paper is mainly focused on Alzheimer's Disease, the title and the objectives should be restricted to AD.
Response: We agreed with the reviewer. The title, second and last lines of the abstract of the review were updated. The objective already states that the review describes CD properties that can lead to the prevention of ND, particularly AD.
7) At the end of the conclusions section, the authors have included some information from the template (This section is not mandatory...).
Response: We appreciate the reviewer's observation. Information from the template was removed.

Reviewer 2 Report
The present review highlighted the role of cardamonin in neurodegenerative diseases, mechanistic insights into cardamonin ability to modify multiple oxidative-stress, antioxidant system pathways, Nrf2, and neuroinflammation. Also, it points to the possible therapeutic potential and preventive utilization of cardamonin in neurodegenerative diseases.
My comments
-The review is confusing in some points. The title, abstract and introduction mention neurodegenerative diseases including Alzheimer’s disease (AD) and Parkinson’s disease (PD) and even introduction gives more examples. however, the rest of review focuses on Alzheimer’s
-7. The Use of Natural Compounds to Prevent Alzheimer's Disease: the revie should focus on cardamonin as indicated in the title. figure 4 include many natural products, why? please be consistent
-As a suggestion, it is better to combine pathological mechanism (Nrf2/NF-kb) in AD followed by the role of cardamonin
-The abbreviations should be carefully used, defined for the first time and then abbreviation should be used consistently, revise for example Alzheimer’s and AD
- Conclusion: "This section is not mandatory but can be added to the manuscript if the discussion is unusually long or complex", please remove
- The review would benefit from summarizing tables
Author Response
Dear Editor:
We are pleased to resubmit the revised version of the Manuscript ID: brainsci-2122210, title: "The Neuroprotective Effects and Therapeutic Potentials of the Chalcone Cardamonin for Neurodegenerative Diseases." We have changed the manuscript and made the requested modifications. We appreciate the reviewers' constructive criticisms and comments and have addressed each of their concerns as outlined below.
Reviewer 2:
1) The review is confusing on some points. The title, abstract, and introduction mention neurodegenerative diseases, including Alzheimer's Disease (AD) and Parkinson's Disease (PD), and even the introduction gives more examples. However, the rest of the review focuses on Alzheimer's.
Response: We agree with the reviewer that some points in the review were confusing. The title, second and last line of the abstract of the review was updated. The purpose of the introduction is to give a general idea about neurodegenerative diseases, which is why we are citing diseases other than AD. However, the objective is clear and already states that the review describes CD properties that can lead to the prevention of ND, in particular AD. We also included two new sections about biomarkers for AD and current FDA-approved medications (Section 5. Biomarkers for Diagnosis and Treatment of Alzheimer's Disease, and Section 6. Current FDA-approved AD medications) to add more information to the review and provide more clarification regarding AD.
2) The use of Natural Compounds to Prevent Alzheimer's Disease: the review should focus on cardamonin as indicated in the title. Figure 4 includes many natural products; why? Please be consistent.
Response: We agree with the reviewer that figure 4 shows many natural compounds other than cardamonin can be confusing. Figure 4 was removed from the manuscript.
3) As a suggestion, combining pathological mechanisms (Nrf2/NF-kb) in AD is better, followed by the role of cardamonin.
Response: We appreciate the reviewer's suggestion. We combined sections 2, 3, and 4 since these sections talked about Alzheimer's Disease, pathogenesis, risk factors, and AD/oxidative stress. We kept sections of Nrf2 and Nfkb in different groups so the reader can have a better understanding of the role of each signaling pathway in inducing or preventing AD. However, when describing the CD effect on these pathways in "Section 7.3.1. Neuroprotective effects of Nrf2 activation," we discuss the two pathological mechanisms together, as they can affect each other.
4) The abbreviations should be carefully used and defined for the first time, and then abbreviations should be used consistently and revised, for example, Alzheimer's and AD.
Response: We appreciate the reviewer's observation, and the whole text was revised to be consistent with the abbreviations. Alzheimer's Disease without abbreviation was kept only on titles.
5) Conclusion: "This section is not mandatory but can be added to the manuscript if the discussion is unusually long or complex" please remove
Response: We appreciate the reviewer's observation. Information from the template was removed.
6) The review would benefit from summarizing tables.
Response: We appreciate the reviewer's suggestion and added a diagram to summarize the data from the review (Figure 4).

Round 2
Reviewer 2 Report
The authors addressed my comments